



# Quantifying local-scale dust emission from the Arabian Red Sea coastal plain

Anatolii Anisimov[1], Weichun Tao[1#], Georgiy Stenchikov[1], Stoitchko Kalenderski[1], P. Jish Prakash[1], Zongliang Yang[2], Mingjie Shi[2*]

[1]King Abdullah University of Science and Technology (KAUST), Physical Science and Engineering Division (PSE), Thuwal, 23955-6900, Saudi Arabia
[2]The University of Texas at Austin, Jackson School of Geosciences, Department of Geological Sciences, Austin, TX 78712, USA
[#]Now at Policy Research Center of Environment and Economy, Ministry of Environmental Protection, Beijing, 100029, People's Republic of China
[*]Now at University of California, Los Angeles, CA 90095, USA

*Correspondence to:* Georgiy Stenchikov (georgiy.stenchikov@kaust.edu.sa)

**Abstract.** Dust plumes emitted from the narrow Arabian Red Sea coastal plain are often observed on satellite images and felt in local population centers. Despite its relatively small area, the coastal plane could be a significant dust source, however, its effect is not well quantified as it is not well approximated in global or even regional models. In addition, because of close proximity to the Red Sea, a significant amount of dust from the coastal areas could be deposited into the Red Sea and serve as a vital component of the nutrient balance of marine ecosystems.

In the current study, we apply the off-line fine-resolution version of the Community Land Model version-4 (CLM4) land surface model to better quantify dust emission from the coastal plain. We verify the spatial and temporal variability of model results using independent station reports. We also compare the results with the MERRA Aerosol Reanalysis (MERRAero) reanalysis. We show that the best results are obtained with 1-km spatial resolution and dust source function based on Meteosat Second Generation Spinning Enhanced Visible and InfraRed Imager (SEVIRI) measurements. We present the dust emission spatial pattern, estimates of seasonal and diurnal variability of dust event frequency and intensity, and discuss the emission regime in the major dust generation hot spot areas. We demonstrate the contrasting seasonal dust cycles in the northern and southern parts of the coastal plain and discuss the physical mechanisms responsible for dust generation.

The total dust emission from the coastal plain appears to be 7.5 Mt per year, with over 65 % of dust emitted from its northern part. The mineralogical composition analysis suggests that the coastal plain generates around 76 Kt of iron oxides and 6 Kt of phosphorus annually. Given the structure of wind circulation in this area and close proximity of the dust hot spots to the sea, we can expect that a significant amount of emitted dust is deposited to the sea, almost matching the annual deposition from major dust storms.



# 1 Introduction

Mineral dust has a significant impact on climate at regional and global scales (Choobari et al., 2014; Knippertz and Stuut, 2014; Shao et al., 2011a). Dust particles also play an important role in soil and forest biogeochemistry. Atmospheric deposition is a vital component of the nutrient balance of marine ecosystems (Jickells et al., 2005; Mahowald et al., 2005; Nickovic et al., 2012 and references therein; Schulz et al., 2012). Dust air-pollution also affects human health, increasing risk for human morbidity and mortality (Knippertz and Stuut, 2014).

The exploration of dust generation and transport, as well as climatology and seasonality of the dust cycle in the Arabian Peninsula, is gaining increased attention in recent years (Hamidi et al., 2013; Hamidi et al., 2014; Kalenderski and Stenchikov, 2016; Kalenderski et al., 2013; Notaro et al., 2013; Notaro et al., 2015; Prakash et al., 2015; Rezazadeh et al., 2013; Shalaby et al., 2015; Shi et al., 2016; Yu et al., 2013; Yu et al., 2015; Alobaidi et al., 2016). Along with a strong climate effect, dust outbreaks in this region affect the nutrient balance of the semi-enclosed Red and Arabian Seas. For example, it was shown that the passage of major dust storms over the Arabian Sea causes chlorophyll blooming (Singh et al., 2008). The Red Sea, bordered by the Sahara and Arabian deserts, and with little or no river discharge and infrequent flash floods from land, is highly oligotrophic, especially in the northern part, rendering nutrients coming from the Indian Ocean almost unobtainable (Acker et al., 2008; Chase et al., 2011; Weikert, 1987). Therefore, atmospheric dust and gaseous depositions are important as nutrient supplies for the Red Sea (Kalenderski et al., 2013; Prakash et al., 2015).

Although previous studies indicate that dust outbreaks are most frequent over the eastern sector of Saudi Arabia (Barkan et al., 2004; Goudie, 2006; Prospero et al., 2002; Shalaby et al., 2015; Washington et al., 2003), satellite images and ground observations show that there is a zone of increased dust activity in the western part of the Arabian Peninsula (Ackerman and Cox, 1989; Furman, 2003; Ginoux et al., 2012; Shao, 2008; Shao et al., 2011a; Walker et al., 2009; Yu et al., 2013). Located next to the Red Sea, the narrow coastal plain could make a significant contribution to the overall amount of dust depositing to the sea, transporting iron, phosphorus, and nitrogen. However, despite the importance of this source area for the nutrient balance of the Red Sea, no specific studies have been focused on the semi-desert coastal region and no estimates of the amount of dust emitted from these areas have been made yet, partly due to the scarcity of observations and partly because the narrow coastal plane is a subgrid area for most global and even regional modeling studies.

The concentration of dust particles in the atmosphere depends on small-scale emission processes, which are spatially heterogeneous and involve complex nonlinear interactions controlled by meteorological conditions and properties of land surfaces. As measurement of emission in field conditions is extremely difficult, numerical models are the principal tools for dust emission evaluation. At the same time, the results from the AeroCom intercomparison project for atmospheric models that comprise aerosol components (Huneeus et al., 2011) suggest large discrepancies in model estimates of global dust emission and deposition up to a factor of 10. Regional uncertainties are probably even higher. Obviously, due to the



relatively small area and complex terrain structure of the western Arabian coastal plain, large-scale and even mesoscale models are not able to reproduce the dust emission processes here with desired accuracy. Even for similar meteorological conditions, a number of studies reported substantial differences in dust fluxes predicted by different models, indicating the model deficiencies in accounting for fine-scale features such as soil texture and surface vegetation cover (Ginoux et al.,

2012; Kang et al., 2011; Koven and Fung, 2008; Prospero et al., 2002; Shao, 2008; Textor et al., 2006; Todd et al., 2008; Zender et al., 2003b). Raupach and Lu (2004) identified key challenges in modeling wind erosion related to the representation of land surface processes, including the fidelity of parameterizations and the availability of high-resolution input data for dust generation calculations. Therefore, to obtain reliable estimates of dust emissions, especially in such highly heterogeneous regions as the Arabian Red Sea coastal plain, fine-resolution surface information is required.

Recently, satellite-derived high resolution datasets of surface properties have emerged and provided an opportunity for improving dust emission calculations (Bullard et al., 2011; Ginoux et al., 2012; Kang et al., 2011; Knippertz and Todd, 2012; Pérez et al., 2011; Shao et al., 2011a). For example, Kim et al. (2013) and Hamidi et al. (2014), using a dynamic vegetation dataset, enabled a simple dust emission scheme to account for the control of seasonally varying vegetation cover on dust emission, which is usually accounted for in more advanced schemes (Bullard et al., 2011; Mahowald et al., 2006; Zender et

al., 2003a). Menut et al. (2013) found that the State Soil Geographic Database (STATSGO-FAO), remapped from the Food and Agriculture Organization of the United Nations (FAO) two-layer 5-minute global soil texture dataset (Nickovic et al., 2012) provides realistic spatial patterns of dust emission for the Middle East and North Africa. Shi et al. (2016) discussed the impact of the satellite-derived vegetation dataset on patterns and intensity of dust emission in the Arabian Peninsula. Many studies have been devoted to accurately locating dust source regions using different criteria, accounting for sediment

availability and erodibility due to geographic influences, and applying satellite datasets to define so-called source functions (Ginoux et al., 2012; Kim et al., 2013; Parajuli et al., 2014; Walker et al., 2009; Zender et al., 2003b).

In this study, we focus on dust emission from a relatively small local area: the narrow semi-desert western coast of the Arabian Peninsula. We employed the high-resolution Community Land Model version-4 (CLM4) with the Dust Entrainment and Deposition (DEAD) module to conduct simulations for the three year span of 2009–2011. Our principle objective was to

obtain new, high-resolution, multi-year estimates of spatial and temporal variability of dust emission and assess the dust's mineralogical composition in order to evaluate its impact on the Red Sea. We utilize the fine-scale input land data derived from satellite-based instruments and examine the model's sensitivity to their horizontal resolution. Using high-frequency satellite measurements, we also calculate and apply the dust emission statistical source function and demonstrate the benefits of using high-resolution inventories of soil characteristics.

We compare the results with independent weather code and visibility reports from meteorological stations. Although these data are indirectly related to local dust emissions and cannot be applied for accurate model validation, they may provide valuable information and serve as a reference for determining optimal model configuration (Engelstaedter et al., 2006;





Tegen, 2003). We also compare (and calibrate) our dust emission estimates with MERRA Aerosol Reanalysis (MERRAero) (Buchard et al., 2016), a recent reanalysis product that includes an aerosol model component and has the highest spatial resolution compared with analogous products.

Marine productivity is largely limited by the availably of iron (Mahowald, 2009), which in turn depends on the solubility of iron-containing compounds in seawater. It has been shown that aerosol source mineralogy is a crucial factor for iron content and solubility as well as aging in the course of particle transport (Baker and Croot, 2010 and references therein). Together with iron, both phosphorus and nitrogen also frequently limit marine productivity (Okin et al., 2011). To evaluate the possible amount of mineral nutrients deposited in the Red Sea, in this study we assess the mineralogical composition of dust emitted from the coastal area, applying the global datasets of soil texture and mineral composition, GMINER30, developed by Nickovic et al. (2012). We assume that the mineral composition and size fractioning of the emitted dust are close to those of the parent soil. This assumption does not always hold (Claquin et al., 1999; Perlwitz et al., 2015). Moreover, airborne dust changes its size distribution and mineralogical composition during its life cycle. Nevertheless, our assessment may serve as an initial estimate of the mineralogical composition of dust particles deposited to the Red Sea, especially because of the short pathway from the coastal plain to the sea, since local dust particles are less processed in the atmosphere than those subjected to long-range transport.

The rest of the article is organized as follows: In Sect. 2, we present the model description and characterize the study domain and observational datasets. In Sect. 3, we describe numerical experiments, examine model sensitivity to land surface datasets and compare results with station observations. A detailed analysis of dust generation and its spatial-temporal variability is conducted in Sect. 4. We summarize our results and draw conclusions in Sect. 5.

## 2 Data and methods

### 2.1 CLM4 model and meteorological forcing

We perform the numerical experiments using the off-line CLM4 (Lawrence et al., 2011; Oleson et al., 2010) implemented with the DEAD module (Zender et al., 2003a). CLM4 is the land surface parameterization model used with the Community Earth System Model (CESM) (Hurrell et al., 2013), and some other regional models [i.e. Regional Climate Model (RegCM4) (Wang et al., 2016) and Weather Research and Forecasting (WRF) (Zhao et al., 2016)]. CLM4 simulates turbulent fluxes of momentum, heat, and water vapor from the surface into the atmosphere, interaction of solar and thermal radiation with soil and vegetation, and heat and moisture fluxes in soils. CLM4 also simulates vegetation processes. The offline version of CLM4 can be run at a finer spatial resolution than driving meteorological fields to account for high heterogeneity of land surface. Additionally, some soil characteristics in CLM4 can be prescribed, instead of being calculated within the model. In




this study, we turn off the transient land cover change calculations and the dynamic global vegetation model to conduct historical simulations using observed high-resolution satellite land cover and vegetation datasets instead.

CLM4 is forced by meteorological fields including wind, surface pressure, precipitation, temperature, and incoming solar and thermal radiation. The driving meteorological fields for CLM4 are provided by the WRF model (Skamarock et al., 2008) run at a 10 km × 10 km resolution over the Arabian Peninsula (8.06° N–34.6° N, 30.3° E–60.9° E) for the period of 2009–2011. The domain completely covers the Arabian Red Sea coastal area (Fig. 1). The WRF configuration used in our simulations is detailed in Table 1.

## 2.2 Dust generation

The DEAD module (Zender et al., 2003a) is designed to calculate dust emission at both local and global scales, generally following the microphysical and micrometeorological model of dust mobilization developed by Marticorena and Bergametti (1995). Soil moisture, vegetation properties, land use, and soil texture data needed to drive DEAD are provided by CLM4. DEAD falls into the category of intermediate complexity models that are more sophisticated than simple bulk mobilization schemes (Tegen and Fung, 1994) and not as complex and calculation-heavy as fully microphysical schemes (Marticorena and Bergametti, 1995; Shao, 2004; Shao et al., 2011b). Intermediate complexity models use microphysical parameterizations where possible, but make simplifying assumptions and use empirical coefficients to shortcut complex calculations (Zender et al., 2003a). The total vertical mass flux of dust $F$ (kg m$^{-2}$ s$^{-1}$), generated from the ground into the atmosphere is calculated using the following equation:

$$F = TSf_m\alpha Q_s \sum_{j=1}^{J} \sum_{i=1}^{I} M_{ij}, \tag{1}$$

where $T$ is a spatially uniform tuning constant that controls the average emission rate (see Sect. 2.5).

The $f_m$ parameter is a grid cell fraction of soils suitable for dust mobilization. It depends on land fraction of bare soil (which is calculated dynamically depending on soil conditions), the plant function type (PFT), leaf area index (LAI), stem area index (SAI), and top soil layer water content, calculated within CLM4.

The $\alpha$ coefficient is a sandblasting mass efficiency that depends on the mass fraction of clay particles (CLY) in the soil, which is defined by SOILPOP30, a 30-second soil population dataset developed by Nickovic et al. (2012) from STATSGO-FAO. This soil dataset is widely used in dust-related studies (e.g., Menut et al., 2013).

$M_{i,j}$ is a mass fraction of each soil source mode $i$, carried in each transport bin $j$. It is calculated based on particle size distribution and mass fraction of each source bin. There are three soil source modes $i$=1,2,3 and four transport bins $j$=1,2,3,4 in CLM4. The transport bins approximate particles with diameters from 0.1 μm to 1 μm, from 1 μm to 2.5 μm, from 2.5 μm to 5.0 μm and from 5 μm to 10 μm. In the original model formulation the dust emission is calculated for each transport bin $j$



separately, summing up the contribution from each soil source mode $i$. Here, we consider total emitted dust mass and therefore, sum up flux into all the transport bins $j$ in equation (1).

$Q_s$ is the total horizontally saltating mass flux (kg m$^{-2}$ s$^{-1}$). It is proportional to the third power of saltation wind friction velocity $u_{*s}$ (m s$^{-1}$) when it exceeds threshold velocity $u_{*t}$:

$$Q_s = \begin{cases} \frac{c_s \rho_{atm} u_{*s}^3}{g} \left(1 - \frac{u_{*t}}{u_{*s}}\right)\left(1 + \frac{u_{*t}}{u_{*s}}\right)^2, & \text{for } u_{*s} > u_{*t}, \\ 0, & \text{for } u_{*s} \leq u_{*t} \end{cases} \tag{2}$$

where $c_s$ is the saltation constant equal to 2.61, $\rho_{atm}$ is the atmospheric density (kg m$^{-3}$), and $g$ is the acceleration of gravity (m s$^{-2}$). Saltation wind friction velocity $u_{*s}$ is calculated from wind friction velocity $u_*$ (m s$^{-1}$) accounting for the Owen effect of increasing $u_*$ during saltation (Zender et al., 2003a). Threshold friction velocity $u_{*t}$ is a function of surface roughness and soil moisture calculated within CLM4.

In the default CLM4 configuration $S = 1$, assuming that the emission is calculated based on winds and available surface and soil properties only. However, it has been reported recently that the models based on purely physical properties of soils represent quite inaccurate spatial patterns of dust emission, especially on the regional scale (Huneeus et al., 2011; Knippertz and Todd, 2012). This is caused by the deficiencies of parameterizations and inaccurate input information. Thus, the spatially varying source function $S$ is introduced to improve the spatial distribution of dust emission simulations. $S$ has a

sense of soil erodibility and accounts for the susceptibility of soil to wind erosion (Webb and Strong, 2011).

Different approaches have been discussed and a number of principles to calculate the source function recently introduced (Kim et al., 2013; Parajuli et al., 2014; Walker et al., 2009). Ginoux et al. (2001) proposed calculating the source function based on a topographic approach, assuming that the areas with topographic depressions are the most probable locations for sediments to accumulate. The geomorphic source function (Zender et al., 2003b) is based on the assumption that dust

emission is likely to occur from areas of potential runoff collection. Similar to the topographic source function, it only depends on elevation. Another family of source functions is instead based on observations (mostly remote sensing), assuming that the most active dust source areas are those where airborne dust is more frequently observed. The statistical source function introduced by Ginoux et al. (2010) and Ginoux et al. (2012) uses MODIS estimates of aerosol optical depth and land cover data to identify the dust source areas.

For the purposes of the current study, we follow the statistical approach by Ginoux et al. (2010) and Ginoux et al. (2012), but calculate the dust source function based on measurements from the Meteosat Second Generation Spinning Enhanced Visible and InfraRed Imager (SEVIRI) instrument. The SEVIRI instrument is located on board the Meteosat-9 geostationary satellite and provides measurements every 15 minutes (Brindley and Russell, 2009; Banks and Brindley, 2013), much more frequently than MODIS. SEVIRI measurements were recently utilized to analyze dust sources in Northern Africa (Evan et





al., 2015). In the present study we use observed atmospheric reflectance, surface albedo, aerosol index, and aerosol optical depth (AOD) to attribute the AOD peaks to dust outbreaks. It is assumed that the intensity of a dust source should be proportional to the frequency of occurrence of atmospheric dust:

$$S = N\,(AOD_D > AOD_t)\,/\,N\,(AOD),\tag{3}$$

where statistical source function $S$ is defined in each location as a number of events when dust-caused $AOD_D$ exceeds the threshold value $AOD_t$, chosen to be 1.12, and normalized to the total number of observations. The threshold value is chosen with respect to the temporal frequency of the SEVIRI instrument. Below we show that the source function based on high frequency measurements significantly improves the simulation results.

## 2.3 Observations, metrics and an overview of the study area

The targeted study area is the eastern coast of the Red Sea in western Saudi Arabia. It is shown in Fig. 1a bounded by a solid red line. The coastal area has the historical name of Tihamah. It covers both plain and hill landscapes, from the Tihamat Al-Hejaz (northern part) and Tihamat 'Asir (southern part) coastal plains to the Scarp Mountains of Midyan, Ash Shifa' and Asir (Edgell, 2006). The land cover, precipitation, and surface wind speed are highly heterogeneous in this narrow (on average 100 km wide) area. In the eastern part of the coastal plain, closer to the mountain area, the land is covered by more or less continuous shrubs and steppe vegetation due to higher precipitation (see Fig. 1c–e). In the northern coastal plain vegetation cover is sparser (Fig. 1f) as the annual precipitation is only 50 mm. Southward, in most of the piedmonts, annual rainfall of 100–200 mm supports denser vegetation cover (Vincent, 2008). Additionally, violent flash floods in this region can add to the complexity by changing the availability of sediments and vegetation cover from year to year.

The Red Sea environment has been identified as a zone of complex wind circulation (Langodan et al., 2014). Due to the strong land-sea diurnal temperature contrasts, land and sea breezes persist through the entire year. The large-scale circulation systems interact with breezes and are reinforced by orographic structures, which creates a complex pattern of mesoscale circulation. The most prominent mesoscale feature of the Red Sea is the Tokar Gap jet on the western coast (Davis et al., 2015 and references therin). Westward-blowing mesoscale jets also exist on the eastern coast (Gille and Llewellyn Smith, 2014; Jiang et al., 2009). These jets originate mostly in Winter due to the cold/dry air outbreaks from the central Arabian plateau and channel through a series of mountain gaps. They may last for several days and have a prominent diurnal cycle. The jets, along with the breezes, cause small-scale dust updrafts in the coastal area. The generated dust plumes are sometimes observed by satellites over the Red Sea. For example, a dust storm with narrow dust plumes caused by the jet winds captured by MODIS/TERRA at 7:45 UTC on 14 January 2009 is shown in Fig. 1b.

In order to cover the study area, we run the CLM4 model over the two rectangular domains shown in Fig. 1a. Also shown are the meteorological observation stations that are used in the current study. We use hourly data from the Integrated Surface Dataset (ISD) developed by the National Climatic Data Center (NCDC) (Smith et al., 2011). We selected 15 stations in



Saudi Arabia and 1 station in Jordan inside the CLM4 domains with continuous observation records for 2009–2011. The stations provide meteorological observations including weather code and visibility reports. The automated visibility measurement and manned weather code observation are reported on an hourly basis, but the weather code is only present when visibility reduces to below 10000 m. Otherwise, just a constant visibility of 10000 m is reported (indicating fair weather). The weather codes that correspond to the presence of dust are: 06 – dust in suspension, 07 – dust raised, 08 – dust whirl, 09, 30 to 35 – dust storm. Most of the weather stations (except that in Makkah) are located on the site of regional or international airports, thus the data archive was primarily assembled from SYNOP or METAR/SPECI weather reports (Smith et al., 2011).

Although the station visibility measurements are only indirectly related to the amount of locally emitted dust, they are one of the most relevant data sources for assessing dust emission fluxes in the absence of other observations. These data are frequently used in dust-related studies. For example, the present weather code reports from meteorological records have been used for evaluation of dust event frequency and dust climatology (Cowie et al., 2014; Goudie and Middleton, 1992; Hamidi et al., 2014; Notaro et al., 2013; Shao and Dong, 2006; Wang et al., 2011; Yu et al., 2013). In some other studies, these observations were used to derive soil erodibility fields (Shao, 2008). The parameterization formula for assessing near-surface dust concentration based on visibility measurement has also been proposed (Camino et al., 2015 and references therein; Rezazadeh et al., 2013; Shao et al., 2003). Mahowald et al. (2007) used the station visibility measurements to study dust sources and stated that visibility-derived observations should better capture temporal variability of surface dust fluxes compared to AOD measurements. But still, these data cannot serve as a quantitative measure of model performance, being non-automated (in the case of weather code), and being highly influenced by remote dust transport, the presence of water vapor, and dust physical properties and composition (Shao, 2008). Another limitation of station observations is a weak sensitivity to low and moderate reductions of visibility that is only reported and complemented by the weather code when it drops below 10000 m. Camino et al. (2015) also note that clear skies are often reported under hazy atmospheric conditions when dust is present. Thus we do not expect these correlations to give an absolute assessment of model emissions, but to allow comparison of different model configurations.

We apply several metrics to compare the model statistics of dust events with station data, making use of both weather code reports and visibility measurements. First, we assess the temporal variability of dust event frequency and intensity, correlating the monthly-averaged time series. We follow the classical definition of dust event frequency $F_d$ from hourly weather code reports (Shao and Dong, 2006):

$$F_d = N_d / N_{tot}, \tag{4}$$

where $N_d$ is the number of reported dust events, and $N_{tot}$ is the total number of reports (including those when visibility was not reduced below 10000 m and no weather code was reported). All of the weather codes indicating the presence of dust (i.e. 06 to 09, and 30 to 35) were considered corresponding to a dust event. Based on this definition, we construct the monthly-





averaged time series, so that the frequency is calculated separately for each month. To obtain the model estimate of dust event frequency, we calculate it as a fraction of time when hourly-averaged emission is above the certain threshold. We apply two constant thresholds of 1 μg m$^{-2}$ s$^{-1}$ and 4 μg m$^{-2}$ s$^{-1}$, approximately corresponding to 70[th] and 85[th] quantiles of hourly emission rates. Taking the fraction of time with dust emission above the threshold during the month, we obtain the

model monthly time series of dust event frequency.

To analyze the intensity of individual dust events, we sample the visibility measurements for each station taking only those time steps that correspond to dust events, and calculate the monthly-averaged visibility reduction, treating it like "dust event intensity". In the case of no dust reports during the month (which is not a rare case for some stations), the 10000 m visibility value is presumed. The corresponding model time series are obtained in a way similar to that of frequency, applying the

same thresholds. Dust generation intensity is considered equal to zero if there are no events above the threshold during the current month. An approach alternative to sampling was proposed in (Mahowald et al., 2007). The authors noted the scarceness of weather code reports and proposed to filter non-aerosol (fog-driven) visibility reductions based on dew-point temperature measurements. In our case, we prefer a sampling approach as most of the station visibility measurements are complemented with weather codes.

Both of the metrics described above reflect primarily temporal, not spatial, variability of the model results. We apply the metrics to different model configurations and, as their basic effect is aimed at improving the spatial patterns, no significant differences are found. Thus some other metrics are needed to assess the reliability of dust emission spatial distributions. The technique we propose for assessing spatial patterns of dust emission is to sample the hourly visibility time series by dust event reports, choosing the time steps when a dust event was reported, and to calculate the daily, and then 3-year mean

visibility for each station. Mean emission rate is also calculated from model data sampled for the same time steps. Station data are sampled to correspond to hourly instantaneous model output: thus SPECI reports that usually take place between regular reports are not considered. We therefore obtain two samples of 3-year averaged station dust intensity and model emission rate (with the sample length equal to the number of stations) and calculate the correlation coefficient between them.

We calculate correlation coefficients between samples that reflect diverse highly non-linear physical phenomena. As we do

not have the physical ground to assume the linear relation between these phenomena, we use Spearman's rank correlations instead of Pearson's correlations for all cases. The dust emissions and station visibility are negatively correlated, whereas the opposite is true of station dust frequency. For the sake of simplicity, here we report the emission – intensity correlations with reversed sign, keeping both coefficients positive.

**2.4 MERRAero reanalysis**

Very recently, a few aerosol reanalysis products have become available (Buchard et al., 2016; Inness et al., 2013). In this study, we utilize the dust emissions from MERRAero aerosol reanalysis developed by NASA (Buchard et al., 2016), which



was calculated using meteorological fields from Modern-Era Retrospective Analysis for Research and Applications (MERRA I) (Rienecker et al., 2011). The reanalysis has a spatial resolution of 50 × 50 km and is available from 2003 onwards. MERRAero is built on the Goddard Earth Observing System version 5 (GEOS-5) atmospheric model, which comprises an aerosol module based on a version of the Goddard Chemistry, Aerosol, Radiation, and Transport (GOCART)

model (Chin et al., 2002; Ginoux et al., 2001). GOCART simulates the interactive cycle of dust, sulfate, sea salt, black and organic carbon aerosols. MERRAero assimilates AOD observations from the MODIS sensor flying both on TERRA and AQUA satellites. The GOCART dust scheme in GEOS5 uses a topographic source function to correct the dust emission spatial distribution.

It is difficult to expect a global reanalysis with a relatively low spatial resolution to produce detailed estimates of dust

emission over a narrow coastal zone. The coastal plain is only covered by one or two grid boxes (in width) by the MERRAero grid. On the other hand, the reanalysis captures the enhanced dust activity area in the western coast of the Arabian Peninsula and its integral (over the entire coastal area) dust emission estimates might be a good first approximation.

We also examined the Monitoring Atmospheric Composition and Climate (MACC) reanalysis product available from ECMWF (Cuevas et al., 2015; Bellouin et al., 2013), but its spatial resolution of 80 × 80 km is coarser than that in

MERRAero and the enhanced dust emission from the coastal plain is not captured.

## 2.5 Dust emission calibration

It is a common approach for atmospheric dust calculations to be calibrated based on observations of total AOD (or assimilate AODs as in MERRAero reanalysis), as it is the basic observed quantity relevant for atmospheric aerosol simulations (Kalenderski et al., 2013; Prakash et al., 2015; Zhao et al., 2013; Zhao et al., 2010). However, in our off-line simulations we

do not calculate AOD, and therefore cannot compare our results with the observed AOD directly (Shi et al., 2016).

As there are no available observations of dust emission, no methods for direct dust emission model validation are available (Laurent et al., 2008). Furthermore, no dust-related observations are available in this narrow Red Sea coastal area in general. Therefore, we calibrate the total model emissions using MERRAero reanalysis (see Sect. 2.4), which applies corrections to match observed AOD values. We rely on the MERRAero estimate of 2009–2011 annual dust emission from the coastal plain

(7.5 Mt), and set the $T$ constant to produce the same dust amount in CLM4. The tuning constant is different when we use or do not use the dust emission source function.

## 3 Sensitivity analysis

The dust emission parameterizations calculate dust influx in the atmosphere using meteorological fields, land-surface physical properties, and, sometimes, empirical proxy information about land-surface erodibility, see Equation (1). To





evaluate the sensitivity of the parameterized dust emissions to varying soil and vegetation physical properties, we first analyze the sensitivity of dust emission flux to spatial resolution of land surface characteristics, then apply the dust emission source function and test the results using weather station data.

### 3.1 Sensitivity to the horizontal resolution of surface data

Shi et al. (2016) discussed CLM4 sensitivity to the type and resolution of vegetation datasets for the entire Arabian Peninsula. They quantified the impact of high-resolution surface characteristics derived from MODIS measurements compared to the default ones on dust emission in the Arabian Peninsula. They found that dust emission is most sensitive to surface vegetation, especially in sparsely vegetated areas, which is the case for the western coastal plain. Here, we extend the sensitivity study of Shi et al. (2016) to finer scales, examining the sensitivity to the horizontal resolution of plant function

type (PFT), leaf area index (LAI), stem area index (SAI), and clay mass fraction (CLY) fields. The description of those datasets is given in Table 2.

First, we consider the sensitivity of dust emissions, when changing the spatial resolution of each of the input surface characteristics separately. We perform a control experiment with all of the surface data taken at 10 × 10 km resolution (HighALL), and four additional simulations. In the LowPFT, LowLAI, and LowCLY experiments, we degrade the spatial

resolution of one of the datasets (PFT, LAI, or CLY, respectively) in comparison with HighALL to 50 × 50 km. In the LowALL experiment, the spatial resolution of all of the above characteristics is degraded to 50 × 50 km. Wind forcing and model grid resolution of (10 km × 10 km) are kept the same for all simulations. See the definition of all relevant experiments in Table 3. Spatially uniform tuning constant $T = 0.011$ is used in all experiments (Table 4) based on HighALL calibration. It is important to mention that $T$ does not affect the spatial patterns of emission, which is the primary focus of our attention in

these experiments.

The differences between annual mean dust generation in HighALL and other simulations is depicted in Fig. 2, a–c. Overall, using high-resolution vegetation results in an appreciable increase in total dust generation with comparable contribution from PFT and LAI datasets. The changes are not strictly additive, as the emission process is non-linear, and are spatially non-uniform. The highest differences occur along the coastal and mountain areas with substantial vegetation cover (Fig. 1a).

Total dust emission from the coastal plain in LowALL is around 10 % smaller than in HighALL. The partial differences are smaller: 6 % (LowLAI) and 3 % (LowPFT). In some areas south of the coastal plain, high-resolution plant function type leads to decreased dust emissions. The difference between the HighALL and LowCLY simulation is not shown, as the changes are very small (less than 1 g m$^{-2}$ a$^{-1}$). The likely explanation for the low model sensitivity to soil texture dataset resolution is that its data sources may initially have been based on relatively coarse resolution observations, which have

subsequently been reinterpolated to a finer grid.



To analyze the impact of fine-resolution surface data on dust generation, we perform an additional FineALL experiment with 1 × 1 km model grid and all the input datasets taken at the highest resolution possible (Table 2). The wind forcing is kept at 10-km resolution. The difference of annual dust emission between FineALL and HighALL simulations is shown in Fig. 2d. The results confirm the previous finding. Total dust emission further increases when switching to 1-km resolution, although

the magnitude of these changes is smaller than the difference between HighALL and LowALL (Fig. 2c). Similarly, the changes are mostly associated with the vegetation and confined to the southern part of the coastal plain.

## 3.2 Model test with station data

In this section, we compare the model results with observations at meteorological stations, keeping in mind the limitations of this approach discussed in Sect. 2.3. To obtain the model values at station locations, we use bilinear interpolation from four

surrounding grid points. If bilinear interpolation is not possible (for the coastal stations), the nearest neighbor grid point is used. First, we assess the model's ability to capture the temporal variability of dust generation in the region on monthly scales. The temporal variability of model dust emissions is mostly driven by the wind forcing. As the wind forcing is the same for all experiments, it is not surprising to find that correlation coefficients are similar in all model simulations. Therefore, we show the results for both dust event frequency (Fig. 3a) and intensity (Fig. 3b) based on HighALL simulation

only.

For most of the stations there are positive correlation coefficients of dust event frequency, ranging from 0.3 to 0.7 with the mean value of 0.47 ± 0.15 for 1 μg m$^{-2}$ s$^{-1}$ threshold and 0.52 ± 0.14 for 4 μg m$^{-2}$ s$^{-1}$ threshold. Most of the correlations are statistically significant at the 95 % level, suggesting reasonable model skill. For Jeddah, Bisha and Madinah, correlations become significant when a larger threshold is applied, and for Najran and Abha, only correlations with the smaller threshold

are significant. The intensity correlations are not fully independent from frequency, as average visibility drop is related to the number of dust reports (in the case of severe dust storms, there are usually a number of concurrent reports that increase the frequency estimate). Despite that, we report visibility-based correlations of intensity, as these measurements are used for the spatial metrics. The results are slightly worse than for dust frequency with the mean correlation around 0.4 ± 0.2 for both thresholds, but correlations are still significant for 12 stations out of 16. Overall, the obtained correlations demonstrate a

good model ability to simulate the monthly variations of dust activities.

Our correlations for most of the stations in western Saudi Arabia (Yenbo, Al Wajh, Jeddah, Makkah, Taif, Tabuk, Jizan) are higher than those between monthly satellite AOD and dust reports calculated by Yu et al. (2013). They noted the contrast between lower correlations for stations in the western coastal area compared to the central parts of the Peninsula, and primarily attributed this to lower frequency of station dust reports in the western coastal plain. Low correlations between

station visibility observations and nearby AERONET measurements were also reported in Mahowald et al. (2007). Yu et al. (2013) offered several explanations for this. Mentioning the instrumental shortcomings of satellite sensors over complex





terrain and their low temporal resolution, the authors indicated that a lot of dust is transported to the western coastal area from remote sources at higher altitudes, and therefore not captured by surface stations. Yu et al. (2013) also suggested that low AOD may be observed during the days when a dust event is reported, attributing it to the small spatial scale of dust plumes over complex terrain. Similar arguments were given in Mahowald et al. (2007).

The results obtained in the current study support these ideas and are consistent with the proposed mechanisms. High correlation coefficients suggest that a large number of dust events registered on stations in western Saudi Arabia are driven by relatively moderate local emissions, whereas the high AOD values are to a large extent due to the elevated layers of dust advected from remote regions. These results illustrate the crucial necessity for high resolution modeling tools, as no low-resolution model could provide the required details. On the other hand, we question another idea proposed in Yu et al. (2013)
that the primary aerosol in the coastal area is not dust. It has recently been shown, based on CALIPSO lidar measurements, that dust is the dominant aerosol type over the Arabian Peninsula and the Red Sea (Osipov et al., 2015).

To assess the spatial distribution of simulated dust emissions and choose the best model settings, we use the metrics described in Sect. 2.3. The model and station data are sampled to include the visibility reductions during dust reports only. The number of reported dust events per station during the 3 years considered in the study ranges from less than 100 at two
stations (72 dust reports in Makkah and 64 dust reports Al Wajh) to up to 400. Given this, the two stations with the lowest numbers of observations are excluded from the final spatial analysis. The Al Wajh station is situated just several hundred meters away from the sea shore, and the low number of dust reports may be due to small-scale circulation features. Model and satellite dataset resolution may be not enough to represent the local circulation, surface characteristics and emission rate with desired accuracy. As for the Makkah station, the low number of dust reports may be caused by instrumentation errors
and insufficient quality control; the station is not collocated with the airport and the data are not used in aviation services.

The spatial correlations with station samples are calculated for three basic simulations (LowALL, HighALL and FineALL) with and without the statistical source function (Fig. 3c). The results show that using the SEVIRI source function significantly improves the spatial structure of dust emission. Even our short-sample statistics allow high correlations to be obtained for all of the three basic simulations. Increasing the surface datasets' resolution, together with applying the source
function, leads to increasing the model skill to 0.68, 0.77 and 0.85 depending on the basic simulation. The correlation coefficients for simulations without source functions are not statistically significant. The LowALL correlation is almost zero, whereas correlations in HighALL and FineALL experiments are almost equal. This result is expected, implying high-resolution datasets only add small-scale details that are difficult to capture with a coarse observational network.

Along with the SEVIRI source function, we use several others (see Sect. 2.2). However, topographic, geomorphic and
MODIS-based source functions are all unable to significantly increase the model skill. For topographic and geomorphic source functions this can be explained by the fact that they were developed for large-scale models initially, thus, they are not



expected to work well on regional scales. The MODIS source function, on the other hand, is based on measurements from a polar-orbiting satellite that has low temporal resolution (only two measurements per day), insufficient to capture the local dust phenomena caused primarily by circulations with a prominent diurnal cycle (Kocha et al., 2013; Yu et al., 2013) (see Sect. 4.3).

## 4 Dust emission climatology

The above analysis shows that HighALL and FineALL model configurations with the source function, based on high frequency satellite measurements, provide quite realistic results. Therefore, we further conduct the analysis of dust emission climatology based on the FineALL simulation using the SEVIRI source function and discuss the major dust source areas within the coastal plain, diurnal and seasonal cycles of emission from those areas, as well as their annual mean and variability.

### 4.1 Emissions from the main dust sources

We first address the spatial distribution of dust generating areas (hot spots), and then turn to the temporal variability. To examine the dust generation regime, we discuss the three-year averaged (2009–2011) spatial patterns of total generated dust amount (Fig. 4), dust emission frequency, intensity and maximum emission rate (Fig. 5). Dust emission hot spots are defined as areas where generated dust amount and emission frequency are two times higher, and dust event intensity is 1.5 times higher than domain-averaged values. The location of hot spots are shown by shaded areas on a real color satellite image (Fig. 4c).

To analyze the mechanisms initiating dust generation, we examine the wind forcing and its variability (Fig. 6). We pay special attention to dust generation mechanisms in the hot spot areas. Although the period of three years is quite short to be considered as climatologically representative, it is shown below that dust generation in this area generally has low interannual variability. In the current and subsequent sections, we use the same threshold for frequency and intensity (4 $\mu$g m$^{-2}$ s$^{-1}$) as for calculation of correlation coefficients with observations.

The total dust emission is spatially variable, changing from zero to more than 100 g m$^{-2}$ in some areas (Fig. 4a). Figure 4a and Fig. 5a–b depict a similar pattern, suggesting that the areas with the largest and most frequent dust outbreaks coincide. The dust emission hot spots occupy around 8 % of the total coastal area (Fig. 4c). The zones where the maximum emission rate occurs (Fig. 5c) comply well with the hot spots. Most of the hotspots correspond to lowlands. The hotspots are located not directly near the coastal areas, but rather near the western hillsides of the Hejaz Mountains, in the dry riverbeds ("wadis") where alluvial deposits are available. The primary hot spot zone in the northern part of the study area (SM1, Fig. 5c) spans along the coast between the cities of Yenbo and Umluj. Emission intensity reaches its maximum value here (over 12 $\mu$g m$^{-2}$ s$^{-1}$), and emission frequency is over 0.25. As seen from Fig. 6d, this hot spot is mostly driven by high winds. Dust





event frequency is highly variable here, which is explained by wind forcing variability (Fig. 6e). Dust generation and wind forcing peak in Spring. These hot spot conditions are prevalent in this part of the coastal plain.

The chain of dust hot spots in the southern part of the coastal plain stretches from Makkah to Abha. Three isolated hot spot zones can be identified. The first one lies to the south of Makkah and Taif (SM2, Fig. 5c). The second zone is in the proximity of Al Bahah (SM3). A third small but intensive zone is located on a coast near the city of Al Qunfudhah (SM4). The frequency of dust events is around 0.25 in these southern hot spot areas, and emission intensity reaches more than 10 µg m$^{-2}$ s$^{-1}$.

The SM2 hot spot is driven by moderate winds with considerable intermonth variability, thus the frequency of dust activity changes during the year, having its peak in Summer months. In the rest of the southern hot spots (SM3 and SM4, Fig. 5c), wind activity is weak (Fig. 6d) and dust emission is mainly facilitated by the low erosion threshold and is increased due to source function correction. The intermonth variability of dust emission is relatively low here, and is predominantly driven by dust frequency variations. There are two other smaller, isolated emission zones: a hot spot near the Gulf of Aqaba in the north (SM6, Fig. 5c) and an intensive hot spot area in the south near Jizan (SM5, Fig. 5c).

Dust emission in the large area between 21° N and 24° N is relatively uniform and reaches quite considerable volume. Although there are no major hot spots, this area contributes significantly to the total dust generation, producing around 2 Mt of dust per year. The annual-mean dust frequency is around 0.15 here, and the average dust intensity is 7–9 µg m$^{-2}$ s$^{-1}$, with both of them reaching maximum in Winter. Dust generation shows high intermonth variability, but in contrast to SM1, the variability is mostly caused by variations in dust emission intensity. Examining the wind circulation in this area, we find that the high variability of dust event intensity is caused by high monthly mean values in Winter and early Spring. Dust intensity averaged over this part of the coastal plain reaches 36 µg m$^{-2}$ s$^{-1}$ during a January 2009 dust storm and 28 µg m$^{-2}$ s$^{-1}$ during March 2011. High maximum values of dust intensity are also seen in Fig. 5c. On the other hand, the intermonth variability of dust frequency is relatively low here. Dust outbreaks are driven by short-lived wind gusts, likely to be explained by diurnally varying jet winds (Gille and Llewellyn Smith, 2014; Jiang et al., 2009). This is confirmed by analyzing the sub-month variability of winds (not shown). During January 2009 and March 2011, hourly wind speed variability was two times higher than average, although mean wind speed was only 20 % higher than annual average. Due to the non-linear character of dust generation, these wind gusts may lead to high monthly values of dust generation intensity. Similar processes also occur on the north of the SM1 hot spot.

The annual mean spatial distribution of dust emission in MERRAero for the period of 2003–2015 is depicted in Fig. 4b. Due to its coarse resolution, MERRAero hardly resolves the local-scale emission areas. Nevertheless, the dust generation pattern reasonably complies with the results obtained with the high-resolution model, and features the primary emission zones. Two major emission zones in Fig. 4b can be identified as SM1 and SM2. The latter source area is the strongest, covering large



neighboring territories. The emission zone near Jizan (SM5) is also present in the reanalysis. Overall, the dust emission patterns from CLM4 and independent reanalysis are quite consistent. Below we show that CLM4 dust emission seasonal cycles are consistent with reanalysis as well.

## 4.2. Temporal variability of dust emissions

### 4.2.1 Seasonal cycle of dust emissions

The seasonal and interannual variability of dust storms in the Arabian Peninsula has been extensively discussed in recent studies (Alobaidi et al., 2016; Notaro et al., 2013; Notaro et al., 2015; Rezazadeh et al., 2013; Shalaby et al., 2015; Yu et al., 2013; Yu et al., 2015). Most of the studies report the period of maximum dust activity is from February until July–August, but the peak month varies depending on location and data source (Notaro et al., 2013; Shalaby et al., 2015; Yu et al., 2013). In the north of the Arabian Peninsula, late Winter – early Spring peak is more common, and in the south – south-east desert regions dust activity tends to reach its maximum in Summer. According to Notaro et al. (2013) and references therein, the late Winter – early Spring dust peak in the north-west is due to the cold fronts associated with cyclones from the Mediterranean, whereas the Summer peak in the south is due to diurnal heating, turbulent mixing, and strong Summer Shamal winds (Yu et al., 2015). In this study, we find the seasonality of dust emission from local sources to be quite consistent with previously reported results.

The seasonal cycles (averaged over three years) of total dust generation, monthly mean dust frequency, intensity and monthly maximum emission rate are shown in Fig. 7. The analysis is conducted over the entire coastal domain and separately for its northern and southern parts (separated at 21° N) and hot spot areas. To compare our model results with reanalysis, the corresponding values from MERRAero averaged over 2003–2015 are also plotted together with standard deviation intervals.

The total emission flux (Fig. 7a) exhibits a pronounced seasonal cycle with a dual maximum in March and July and minimum in February and October. The peaks originate from a distinct character of seasonal cycles in the northern and southern parts of the coastal plain. The March peak is only evident in the northern area and is mostly caused by increased intensity during the dust storm episodes (Fig. 7c). High intensity is also seen in January in the north, partially caused by a dust storm in 2009. The peak Winter and Spring seasons for dust intensity in the north are also shown in Fig. 5b. Conversely, the July peak is due to both frequency (Fig. 5a, Fig. 7b) and intensity (Fig. 5b, Fig. 7c) reaching its maximums in the southern part of the coastal plain, although they are lower than that in the northern coastal plain. Seasonal cycle of maximum dust emission rate (Fig. 7d) generally follows that of intensity. Overall, we can conclude that the different climate and surface conditions in the north and south of 21°N drive the spatial variations of the seasonal cycle of dust emission.





The seasonal cycle of dust emissions is dominated by emissions from the hot spots. Since the hot spots are in both the northern and southern parts of the coastal plain, the seasonal cycles of total emissions are smoother than for the northern and southern coastal plain separately. In the hot spots, magnitudes of dust frequency, intensity and maximum emission rate are 2–2.5 times higher than that for the total coastal area and are above the mean plus standard deviation threshold in

MERRAero. The overall amount of dust emitted from the hot spot areas is 1.9 Mt $a^{-1}$ or 25 % of the total emissions, while hot spots occupy only 12800 $km^2$ or less than 10 % of the total area. This fact indicates that the soil mineralogical composition and wind variability have to first be studied in these hot spot areas (Prakash et al., 2016).

The seasonal cycles of dust emissions in MERRAero and CLM4 show similar behavior. As CLM4 dust emissions are scaled to match MERRAero, we only compare the seasonal variations, not averages. In general, seasonal cycles are in good

agreement. Summer dust emissions are the largest in MERRAero, similar to CLM4 results in the southern part of the coastal plain. The Spring peak is not present in reanalysis. One of the possible reasons is the coarse resolution of reanalysis that does not capture the local-scale wind patterns that cause the Spring peak. Similarly, Yu et al. (2013) reported that satellite AOD measurements in the western Arabian Peninsula do not feature the early Spring peak (as opposed to station dust records), attributing it to the local character of springtime dust generation.

With the exception of the March peak, the seasonal cycle of CLM4 dust generation lies within the MERRAero standard deviation interval. This is also true for dust event frequency and intensity, although frequency is slightly smaller than in reanalysis and intensity is slightly larger. It may be caused by the fact that, in the case of MERRAero, these quantities were calculated with the same threshold, but based on three-hourly data; therefore, some dust outbreaks on the threshold borderline are missed. As expected, maximum dust emission rates in CLM4 are larger than in reanalysis, being substantially

above the standard deviation interval, especially in March and July.

The annual dust emission from the entire 147000 $km^2$ coastal area scaled to match MERRAero reanalysis is 7.5 Mt. Emission from the northern region is about 4.9 Mt $a^{-1}$, or 65 % of the total emission. The average dust emission for the total 2003 – 2015 period covered by MERRAero is 7.2 ± 0.5 Mt $a^{-1}$. Small inter-annual variability of emission in reanalysis confirms that the study area is a stable dust source. This estimate of dust generation is comparable with the estimate of

annual dust deposition to the Red Sea due to the major dust storm of 6 Mt $a^{-1}$ (Prakash et al., 2015). Therefore, stable coastal dust sources could also provide a significant amount of mineral nutrients for the Red Sea.

### 4.2.2 Diurnal cycle of dust emissions

The annual average diurnal cycles of total dust generation, frequency, intensity and maximum emission rate are computed from the 3-year simulations (Fig. 8 a–d). All of the quantities have a pronounced diurnal cycle, consistent with wind speed

intensifying during solar peak. Both total dust emission and frequency peak around late afternoon, at 12:00−14:00 UTC, with a slight shift between the northern and southern parts of the coastal plain due to latitudinal extent. The frequency of dust





events during the daily maximum is around 0.35 both in the north and in the south. Overall, around 80 % of airborne dust is generated between 07:00–16.00 UTC. The nighttime dust emission in the northern part is much stronger due to the larger number of cold fronts passing through the northern Red Sea (Notaro et al., 2013; Yu et al., 2015). In the south, the frequency of nighttime dust events is lower due to the different character of wind forcing with a more pronounced diurnal cycle (Notaro et al., 2013). The frequency of dust events in the hotspot areas during the peak hours reaches 0.8, but during the nighttime it is less than 0.05.

Dust emission intensity has a different diurnal cycle. In the north, the daytime maximum of dust frequency corresponds to minimum intensity. The total distribution of dust events above the threshold during these hours is characterized by the large number of moderate-intensity events, thus, the average emission is relatively small. On the other hand, the small total number of dust events above the threshold in the nighttime leads to a larger contribution from strong events and increased average intensity. The diurnal range of emissions in the northern coastal plain is from 7 to 12 $\mu g\ m^{-2}\ s^{-1}$. In the southern part, the nighttime intensity is smaller due to the presence of areas with zero contribution to the average intensity, as there are no dust events exceeding the threshold intensity. This results in an almost uniform diurnal intensity cycle in the southern part of the coastal plain (5–7 $\mu g\ m^{-2}\ s^{-1}$). In the hot spot areas, average dust intensity has two diurnal peaks at 13.00 and 22.00, and reaches minimum at 17.00 UTC. The dust maximum emission rate also has different diurnal cycles in the north and south. It peaks at 8:00 UTC in the northern areas with a diurnal range of 20–50 $\mu g\ m^{-2}\ s^{-1}$, and at 13:00 UTC in the south with a diurnal range of 15–35 $\mu g\ m^{-2}\ s^{-1}$. In the hot spot areas, daily maximum emission reaches 100 $\mu g\ m^{-2}\ s^{-1}$ at 12:00–13:00 UTC.

### 4.3 Mineralogical composition

Dust elemental composition has a variety of physical and biogeochemical impacts. Perlwitz et al. (2015) and Zhang et al. (2015) have applied sophisticated modeling tools to study the dust mineral composition on global scales. In our case, we concentrate on a fine-scale narrow coastal area, as generated dust has the potential to deposit directly to the sea. Thus it is also important to consider its fractional mineral composition. To calculate the emitted mineral fluxes from the Arabian coastal plain we apply the global datasets of dust mineral composition, GMINER30, and soil texture, SOILPOP30, developed by Nickovic et al. (2012). We assume that the relative proportions of minerals in the airborne dust are the same as those of the parent soils, which usually only holds for clay soil fraction (Caquineau et al., 1998; Lafon et al., 2004). The largest size bin of emitted dust (transport bin) in CLM4 is 5–10 μm, while the silt fraction in GMINER30 corresponds to 2–50 μm, which allows us to assume, following Nickovic et al. (2013), that emitted dust is a mixture of clay and silt particles only (without coarser fractions). We must stress that these strong assumptions only allow for a rough estimation of the amount of minerals deposited to the sea, as chemical and physical fractionation processes that change the size distribution and mineral composition of dust occur during both atmospheric transport and deposition.



The emitted mineral fractions are weighted with the clay and silt content in the soil. For minerals that are present in both clay and silt, the weighted values are summed. The minerals' annual emissions are calculated using dust emission flux obtained with the FineALL simulation and the SEVIRI source function applied. Figure 9 shows annual amounts of minerals emitted from the coastal plain. Quartz is the most abundant mineral, comprising around 40 % of the total emission. 25 % of the total

emission corresponds to feldspars, followed by illite, smectite, kaolinite, calcite, gypsum, hematite, goethite (the iron source), and phosphorus. The Arabian Red Sea coast provides 76 Kt of iron oxides and 6 Kt of phosphorus annually. Over 60 % of iron oxides and phosphorus are emitted from the northern part of the coastal plain, acting as a nutrition source for the oligotrophic northern part of the Red Sea

Although only a portion of dust emitted from the Arabian coast is deposited to the Red Sea, due to the close proximity of the

dust generation area to the sea (especially in the northern coastal plain) and the structure of mesoscale circulation that includes jets and breezes, its role in total mineral deposition to the Red Sea could be significant.

**5 Conclusions**

This study focused on the dust emission from the Red Sea Arabian coastal plain. We applied the off line CLM4 land surface model to perform high-resolution simulations of dust emission for 2009–2011 using up-to-date land surface datasets. The

magnitude of dust emission was tuned to fit the estimate from MERRAero reanalysis, while the spatial structure was calculated within CLM4, forced by $10 \times 10$ km resolution meteorology from WRF simulations.

An alternative to off line dust emission simulations could be coupled dust and meteorology calculations within, e.g., the WRF-Chem model (Grell et al., 2005). WRF-Chem includes interactive calculations of transport and chemical/microphysical transformations of trace gases and aerosols, including mineral dust. However, the WRF-Chem model is computationally

demanding and at present cannot be used for multi-year fine resolution simulations in a meaningfully large spatial domain.

To test the simulated dust emission, we developed the corresponding metrics and performed a comparison with the weather station reports of horizontal visibility and present weather code. We obtained significant correlations for monthly time series of dust event frequency and intensity (station-mean correlation coefficients of 0.5 and 0.4), indicating reasonable model performance. The results confirmed that dust emission from local sources on the Arabian Red Sea coastal plain is significant

and supported the hypothesis by Yu et al. (2013) that the dust activity in this area may be caused by local-scale dust outbreaks.

Within the proposed framework, we performed a sensitivity study and demonstrated the importance of using high-resolution input surface datasets. The spatial resolution of vegetation datasets was shown to alter total dust emissions by up to 10 %. We confirmed the conclusions drawn by Shi et al. (2016), showing that the increased resolution of the vegetation dataset



leads to significant dust flux in some zones where it was very weak, when coarse input data fields were used. We estimated comparable contribution to total dust emission from the increased resolution of the plant function type dataset on one hand, and the leaf area index and stem area index on the other hand.

To improve the spatial structure of dust generation, we applied a statistical source function based on the high-frequency geostationary measurements from the SEVIRI instrument. We showed that this approach allows a better representation of dust sources. Depending on model resolution, the statistically significant model skill (spatial correlation coefficient based on comparison with 14 ground stations) varied from 0.68 to 0.85. Without the source function, the spatial model skill was not statistically significant.

Following the model evaluation tests, we based our estimates on model simulation with $1 \times 1$ km spatial resolution and SEVIRI source function. The estimate of total dust emission from the coastal plain, tuned to fit emissions in the MERRAero reanalysis, is 7.5 Mt a$^{-1}$ (approximately 50 g m$^{-2}$ a$^{-1}$). Over 65 % of dust is generated in the northern part of the coastal plain. The seasonality of dust emission differs substantially in the northern and southern parts of the coastal plain. In the south, the annual maximum of dust emission occurs in July, whereas in the north March is the peak month of dust activity. This distinct character is due to the contrasting forcing mechanisms: in the north, emission is caused by strong, diurnally variable, cold season winds, whereas in the south it is largely controlled by a low erodibility threshold and soil moisture. These features result in dual maximum values within the seasonal cycle of total dust emission from the coastal area in March and July.

The spatial pattern of total dust emission is highly non-uniform, reaching more than 100 g m$^{-2}$ in some hot spot areas. The chain of hot spots stretches alongside the coastal zone, with most of them located in the lowlands near the western hillsides of the Hejaz Mountains – riverbeds that are usually considered the source of alluvial material. The hot spots occupy around 8 % of the coastal area and generate over 25 % (1.9 Mt a$^{-1}$) of total dust. The emission pattern is in reasonable agreement with coarse-scale results from the MERRAero global reanalysis, despite the fact that the reanalysis dust model uses a different source function. We also showed that dust generation has a pronounced diurnal cycle. Around 80 % of dust is generated during daytime, between 07:00–16.00 UTC, with dust emission rate and emission frequency peaks during late afternoon (12:00–14:00 UTC).

To assess the mineralogical composition of dust emission, we applied the dataset of soil mineralogy and texture. According to the results, 76 Kt of iron oxides and 6 Kt of phosphorus are emitted from the coastal plain annually. The intensive northern dust hot spot areas are located in close proximity to the Red Sea, and mesoscale jets and breezes could act as efficient mechanisms for bringing the emitted dust to the sea surface. The 7.5 Mt a$^{-1}$ estimate of total dust generation is comparable to the estimate of annual dust deposition to the Red Sea of 6 Mt a$^{-1}$ due to major dust storms (Prakash et al., 2015). Therefore, we expect that the coastal plain could be an essential source of minerals for supporting the nutrient balance of the Red Sea.





**Author contributions**

Anatolii Anisimov performed the data processing, developed the technique for comparison with observations, conducted the comparison with reanalysis, formulated the results and wrote the final paper.

Weichun Tao designed the experiments, ran the model simulations, calculated the source functions, performed the basic

analysis and wrote the paper draft.

Georgiy Stenchikov formulated the problem, directed the research, and edited the paper.

Stoitchko Kalenderski ran the WRF model to obtain the meteorological forcing for the dust emission calculations.

P. Jish Prakash and Weichun Tao worked together on the dust mineral analysis.

Zongliang Yang, one of the developers of the CLM4, helped in setting the CLM4 runs.

Mingjie Shi and Weichun Tao worked together on collecting the land surface data.

**Acknowledgements**

We thank Prof. V. Ramaswamy and Dr. Paul A. Ginoux of GFDL for valuable discussions. We also thank Ms. Linda Everett for proofreading the article. The research reported in this publication was supported by the King Abdullah University of Science and Technology (KAUST). For computer time, this research used the resources of the Supercomputing Laboratory

at KAUST in Thuwal, Saudi Arabia.

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



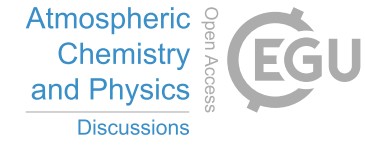

Table 1. WRF model configuration.

| Process | WRF Option |
|---|---|
| Microphysics | Lin |
| Shortwave radiation | Goddard |
| Longwave radiation | RRTM |
| Cumulus parameterization | Kain-Fritsch |
| Surface layer | Monin-Obukhov |
| Land-surface model | Noah LSM |
| Boundary layer scheme | YSU |
| Boundary and initial conditions | NCEP Final Analysis FNL |
| Sea Surface Temperature | NCEP RTG_SST_HR |



Table 2. Land surface data used in model setup.

| Input data | Parameters affected | Default data in CLM4 | Data used | |
|---|---|---|---|---|
| | | Resolution | Original data Resolution | Source |
| PFT | | | 500m × 500m | MODIS Land Cover Product MYD12 |
| LAI | $f_m$ | | 1km × 1km | MODIS MCD15 |
| SAI | | 0.5° × 0.5° | 1km × 1km | Calculated from LAI |
| CLY | $\alpha$, $f_w$ | | 1km × 1km | STATSGO-FAO (10km × 10km) |
| ERD | S | Constant=1 | See Table 4 | |





Table 3. Spatial resolution of input datasets used in simulations.

| | | Simulation | | | | | |
|---|---|---|---|---|---|---|---|
| | | HighALL | LowALL | FineALL | LowPFT | LowLAI | LowCLY |
| Input data | PFT | 10 km | 50 km | 1 km | 50 km | 10 km | 10 km |
| | LAI & SAI | 10 km | 50 km | 1 km | 10 km | 50 km | 10 km |
| | CLY | 10 km | 50 km | 1 km | 10 km | 10 km | 50 km |
| | Wind forcing | 10 km | | | | | |



Table 4. Tuning constants used in the simulations

| Source function | Algorithm | T in (1) | Data source | Remarks |
|---|---|---|---|---|
| No source function | Eq. (1), $S = 1$ | 0.011 | Calculated based on HighALL experiment | Used in LowALL, LowPFT, LowLAI, LowCLY, HighALL and FineALL experiments. |
| SEVIRI statistical | Eq. (3) | 1.28 | SEVIRI AOD data (Brindley and Russell, 2009; Banks and Brindley, 2013) | Used in FineALL simulation with SEVIRI source function. |





**Figure 1. (a) The model domain, study area, and 16 ground observation stations. (b) Dust plume above the Red Sea observed by MODIS/TERRA at 7:45 UTC on 14 January 2009. Overview of the landscapes: (c) piedmont; (d) trees over the sand; (e) wild watermelons over the sand; (f) sand dunes and scattered vegetation.**





**Figure 2.** Differences between annual mean dust emission in model simulations (g m$^{-2}$ a$^{-1}$): (a) HighALL − LowLAI; (b) HighALL − LowPFT; (c) HighALL − LowALL; (d) FineALL − HighALL.





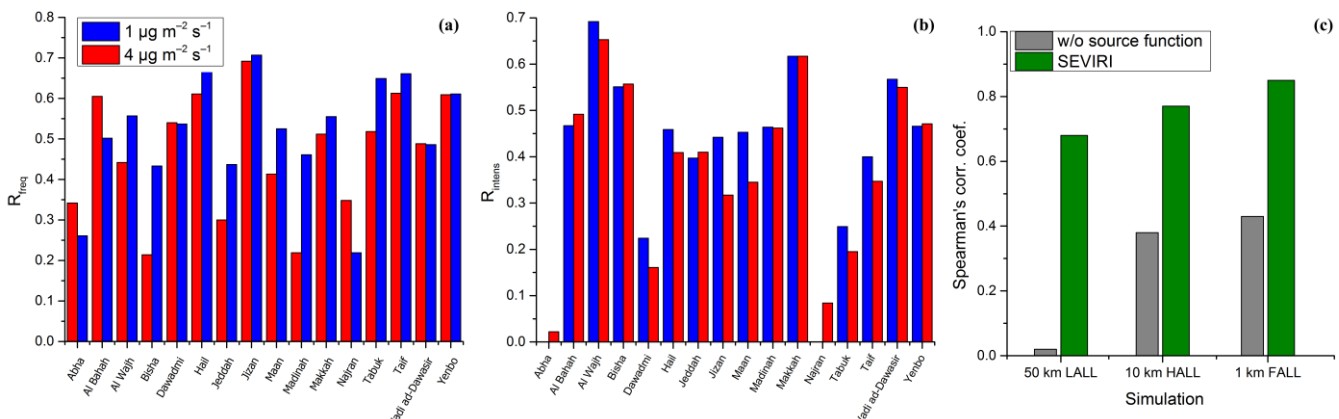

**Figure 3. Spearman's correlation coefficients for monthly-mean series of (a) dust event frequency and (b) intensity between station data and results from HighALL experiment. (c) Spatial metrics of model performance (see text for definition) for three basic experiments with and without SEVIRI source function.**



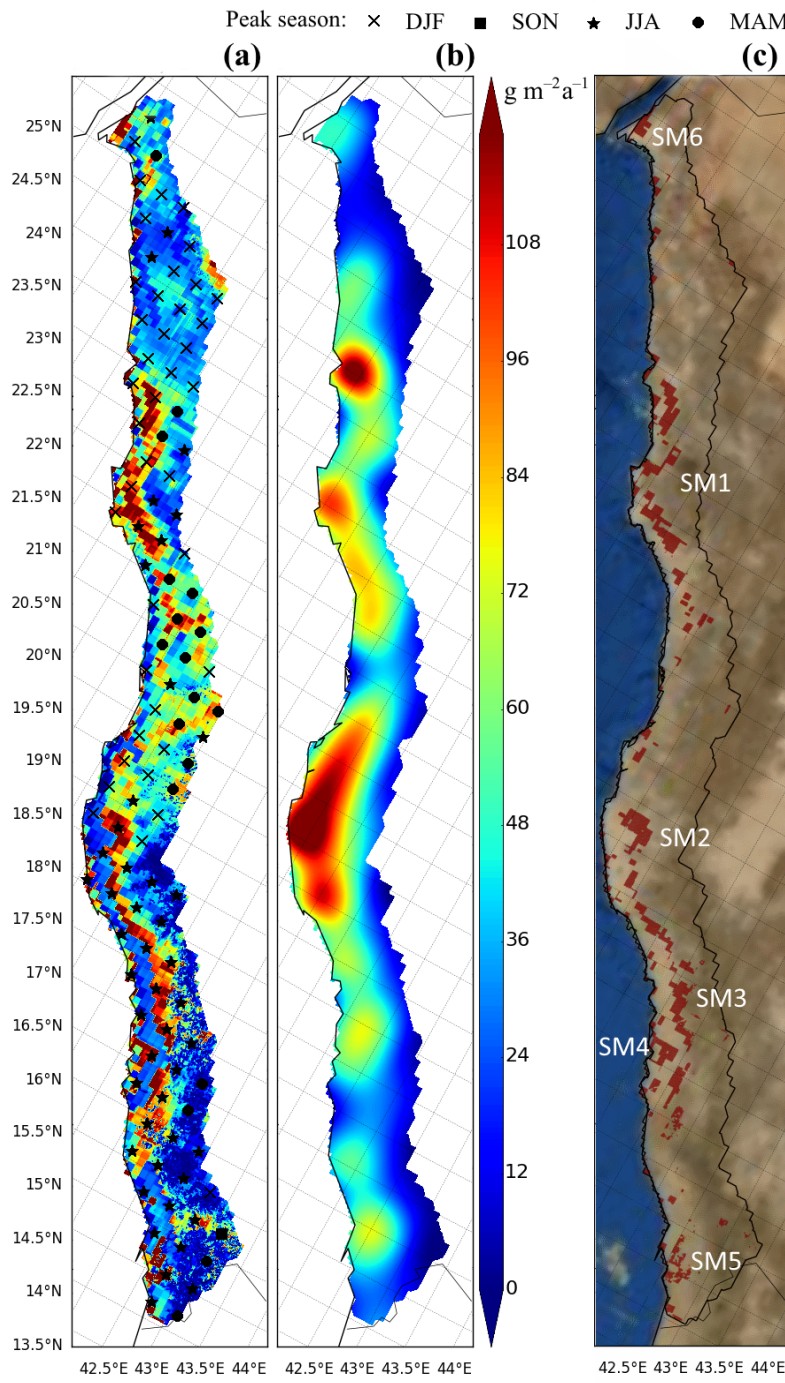

**Figure 4. Annual dust emission (g m⁻² a⁻¹) in (a) FineALL experiment with SEVIRI source function (2009–2011); (b) MERRAero reanalysis (2003–2015). (c) Main dust emission hot spot areas mapped on real color satellite image. Peak season is shown by marks (see figure legend).**







**Figure 5.** Average 2009–2011 (a) dust event frequency; (b) average emission intensity ($\mu$g m$^{-2}$ s$^{-1}$); (c) yearly maximum emission rate ($\mu$g m$^{-2}$ s$^{-1}$) in FineALL experiment with SEVIRI source function. Peak season is shown by marks (see figure legend).





**Figure 6. Standard deviations of monthly (a) total dust emission (g m⁻² mon⁻¹); (b) dust event frequency; (c) average emission intensity (μg m⁻² s⁻¹) in FineALL experiment with SEVIRI source function. Average 2009–2011 WRF forcing (d) wind speed (m s⁻¹), and (e) its monthly standard deviation (m s⁻¹). Peak season is shown by marks (see figure legend).**





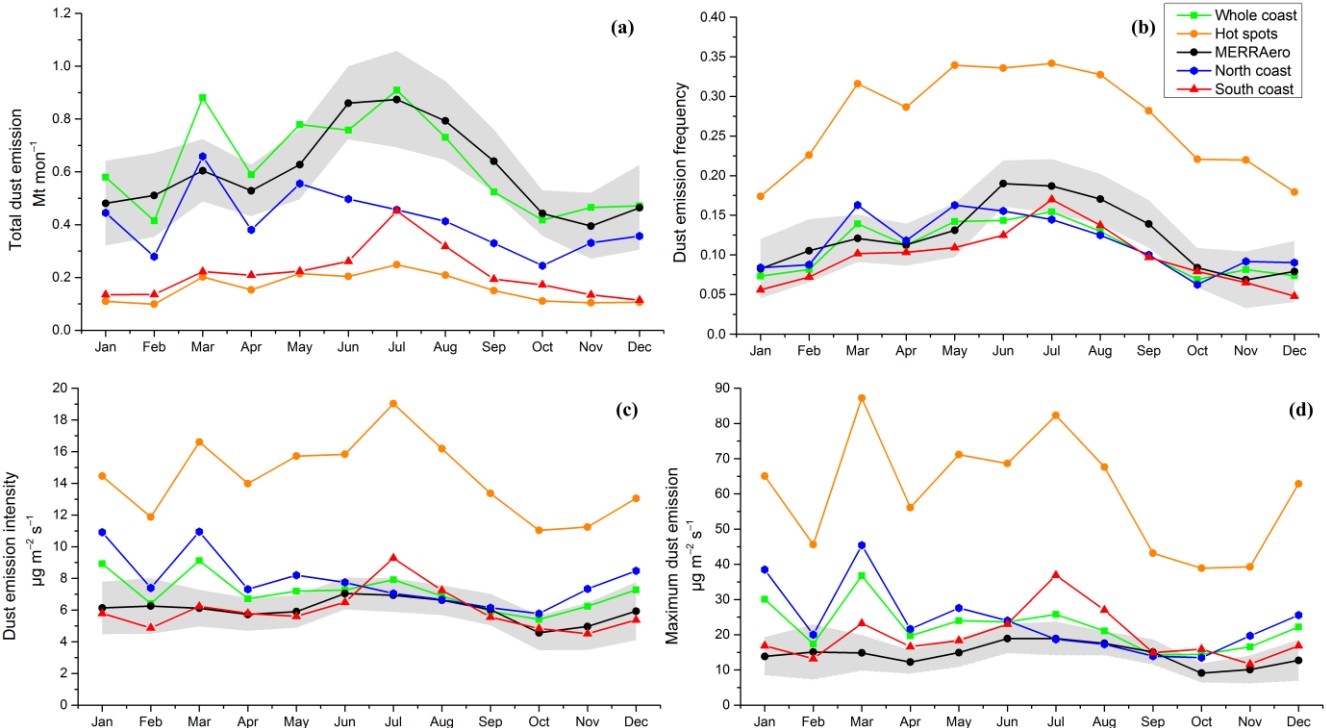

**Figure 7.** Average seasonal cycles of monthly (a) total dust emission (Mt mon$^{-1}$); (b) dust event frequency; (c) average emission intensity (μg m$^{-2}$ s$^{-1}$); (d) maximum emission rate (μg m$^{-2}$ s$^{-1}$) in FineALL experiment with SEVIRI source function (2009–2011) and MERRAero reanalysis (2003–2015). MERRAero Standard deviation intervals are shown by shading.





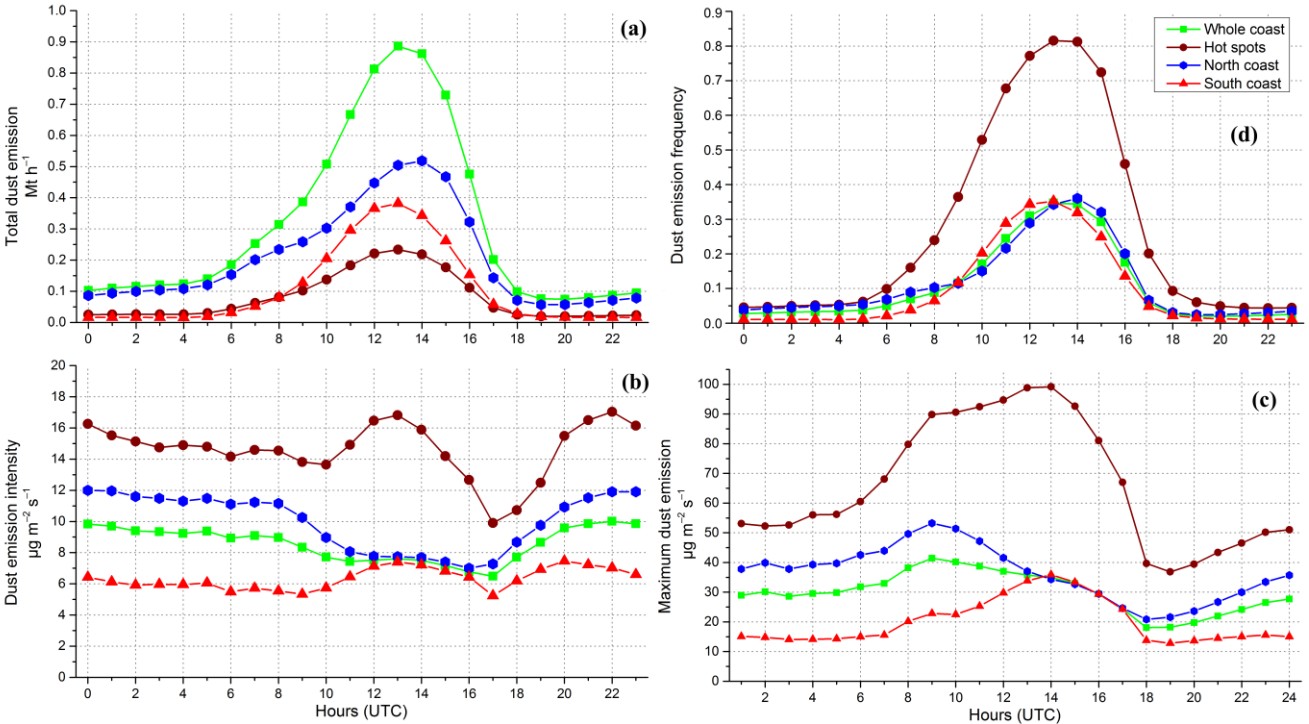

**Figure 8. Annual mean diurnal cycles of (a) total dust emission (Mt h⁻¹); (b) dust event frequency; (c) average emission intensity (µg m⁻² s⁻¹); (d) maximum emission rate (µg m⁻² s⁻¹) in FineALL experiment with SEVIRI source function (2009–2011).**



**Figure 9. Annual mineral emission fluxes (Mt a⁻¹) in FineALL experiment with SEVIRI source function (2009–2011).**