# Peer review of "Quantifying local-scale dust emission from the Arabian Red Sea coastal plain"

_Atmospheric Chemistry and Physics, 2016_

## Referee Comment (RC1) · Anonymous Referee #1 · 20 Sep 2016

The authors study the emission of dust from the Red Sea Arabian Coastal Plain using an off-line land surface model and compare with weather station reports, and attempt to identify emission temporal and spatial patterns and estimate dust composition.

Major revisions would be necessary to reach the scientific and presentation standard of ACP.

Major Comments: =============== Section 2.1: The authors concentrate on a narrow (∼100km) coastal plain with complex circulation (e.g. sea breezes) but use a model resolution that is too coarse (10km). They use the same resolution also when testing varying emission resolutions. Even if this is due to computational constraints, surely short tests of specific dust events can be conducted at higher resolution to properly test dust dispersion. Can the authors identify such events and complement the paper

with higher resolution model runs?

The authors allude to a year-to-year strong variability (Sec.2.3, p.7 l.18), yet only simulate the short period of 3 consecutive years. Are the results meaningful and how are any resulting systematic uncertainties constrained?

The authors do not address dust aging and its potential effects. They should comment if it's relevant before deposition. Does short atmospheric residence time make it not significant?

Section 2.2 describes mostly the dust generation mechanisms that I understand is not the authors own work and is available in the literature. The whole section should be shortened, replacing the details of model parameterisations with references.

Section 2.4,5: Given the very coarse resolution and the relatively small size and high complexity of the region under study of the MERRAero grid, is it advisable to scale to the emission total? What is the overall point of this section? Sections 2.4 and 2.5 should be merged.

Section 3.1: Given the same forcing in all experiments, and differences of a few percent between emission resolutions, what is the point of sensitivity tests? Surely the sensitivity to meteorological conditions and climatic variability is more important and should be studied. The authors should comment and better motivate their study.

Section 3.2: Surely statistical testing can't involve changing thresholds until significance is reached. Also are monthly values used when hourly are available? Given the strong diurnal variability, isn't the latter advisable?

Section 4: Again given varying vegetation and land type, isn't 3 years to short a time span to produce an emission climatology?

Minor Comments: =============== Using the study-specific acronyms FineALL, HighALL, LowALL, etc. is confusing to the reader. Propose to just change with quoting model resolution and if necessary input data (ie. 1km, 10km, 50km, etc.)

[Figure]

Please include the span of years modelled in the abstract.

Abstract l. 26: appears to be -> is estimated to be [...] suggests -> shows

p.9 2nd paragraph: First it's stated that it is not rare to have no dust reports, thus the visibility measurement is used, then that the authors prefer sampling as most station visibility are complemented with weather codes. Aren't these statements contradictory? Please re-write the paragraph for more clarity.

p.9 l.11 current month -> that particular month

p.13 l.9 Please elaborate how Yu et al is questioned based on the results of the present study - not just by quoting another paper.

p. 19 l.8 Paragraph/line incomplete? Missing full stop?

---

## Referee Comment (RC2) · Anonymous Referee #2 · 27 Sep 2016

The manuscript presents an analysis of dust emission from the coastal plain of the Arabian Peninsula at the Red Sea over the time period 2009-2011. This region is an important local dust source with frequent dust events, for which a systematic study of dust emission has been lacking so far. The analysis is done based on off-line simulations with a high resolution land model that includes a dust emission scheme. Thus, feedbacks with meteorological variables are not taken into consideration. The study includes evaluation of the results against observation derived variables. Sensitivity analysis is carried out to study the dependence of the results on the horizontal model resolution, resolution of vegetation and soil data set, and the applied source function. The analysis of dust emission is then done using the model version with $1 \times 1$ km resolution. A total value of the dust emission from the region is provided as well as geographical and temporal patterns, and estimates for the amount of annually emitted

iron oxides and phosphorus.

The study presents new and scientifically interesting results. It is generally well structured and well written. Some points should be addressed before publication, though, which are listed below:

1. **Abstract, main part, and conclusions:** The amount $7.5\,\mathrm{Mt\,a^{-1}}$ of total dust emission as presented in the abstract, the successively derived magnitude of dust emission from different locations, and the quantification of the amount of iron oxides and phosphorus are all predetermined by the calibration of the model. For the lack of measured dust emission rates, the calibration is done by assuming the same emitted dust amount in the land model as in the MERRAero reanalysis over the investigated time period, which is not based on measured values, but calculated using a dust module.

   This approach rests on the assumption that the magnitude of the dust emission in MERRAero is a reliable estimate of the true dust emission from this region. To my knowledge, no evaluation has been published with respect to the dust emission from this region in MERRAero. The authors themselves acknowledge that the resolution of the reanalysis is too low to provide reliable estimates of the dust emission from this region, and they show with their own analysis that the magnitude of dust emission increases with refinement of the horizontal resolution.

   Thus, this suggest the conclusion that the magnitude of the presented total dust emission, the emission amount from individual locations, and the amount of iron oxides and phosphorus are highly uncertain. This uncertainty should be addressed. Perhaps, one could use the variability of the emission in MERRAero from the whole time period 2003-2015, which already has been used in the manuscript by the authors, to provide a first estimate of the lower and upper range of the emission related quantities presented in the paper, especially for

the ones presented in the abstract, even though that still wouldn't address any possible bias in the MERRAero dust emission. The issue of the uncertainty and its sources for the estimates provided in the paper should also be thoroughly discussed in the conclusions. Also, when providing the absolute quantities in the abstract, it should be pointed out that the values are just first estimates that are still highly uncertain.

2. **Page 4, lines 10–13:** The assumption that the mineral composition of dust aerosols and the mineral composition of the soils in *Claquin et al.* (1999) and *Nickovic et al.* (2012) were close does not (always) hold just because of changes during the life cycle of dust from emission to deposition, even more importantly, it does not hold because the measurements of the soil mineral fractions were done for soils that had been wet-sieved. Wet sieving is a technique that strongly disperses soil aggregates (*Shao*, 2001), which is not realistic for dust emission from the parent soils of the dust sources. This caveat to the assumption made by the authors should be added to the manuscript.

Having said this, the authors are mainly interested in the amount of iron oxides and phosphorus. *Nickovic et al.* assume the same iron oxide fraction in the clay and silt size range, and phosphorus is provided only for the clay and the silt-size range together. Therefore, the fractions of these minerals are less affected by the wet-sieving problem, based on these assumptions. Also, in the present manuscript, only the integrated amount over all size bins defined in the dust module is presented. Thus, other sources of uncertainty probably affect the calculated iron oxide and phosphorus amount more than the wet-sieving problem. The wet-sieving issue still may be are more relevant source of error for the other minerals presented in Figure. 9, though.

3. **Page 7, lines 4–7:** How the choice for the threshold value for the statistical source function was made should be explained in detail. It is not clear for the

reader from simply stating, "The threshold value is chosen with respect to the temporal frequency of the SEVIRI instrument".

4. **Page 15, line 5:** Do not say "Dust emission climatology", since the analysis is done only for three simulates years. Name the section "Multi-year dust emission" or similar.

5. **Page 14, line 23:** The unit of the total dust emission in the text should be the same as in Fig. 4a, i.e., $\mathrm{g\,m^{-2}\,a^{-1}}$.

6. **Page 17, line 29:** "All of the quantities have a pronounced diurnal cycle, ..." should be phrased more precisely as "Total dust generation, frequency, and maximum emission rate have a pronounced diurnal cycle, ...". The authors themselves discuss the exception for the intensity further below.

7. **Page 18, line 20:** Add *Scanza et al.* (2015).

8. **Page 31, Table 1:** The used individual components of the WRF model configuration should be presented in a way that is friendly to the reader who is not an insider of the WRF model. That is, not just by using acronyms, but fully spelled out, with references added and information how these components can be accessed.

**References**

Claquin, T., M. Schulz, and Y. J. Balkanski (1999), Modeling the mineralogy of atmospheric dust sources, *J. Geophys. Res.*, *104*(D18), 22,243–22,256, doi:10.1029/1999JD900416.

Nickovic, S., A. Vukovic, M. Vujadinovic, V. Djurdjevic, and G. Pejanovic (2012), Technical Note: High-resolution mineralogical database of dust-productive soils for atmospheric dust modeling, *Atmos. Chem. Phys.*, *12*(2), 845–855, doi:10.5194/acp-12-845-2012.

Scanza, R. A., N. Mahowald, S. Ghan, C. S. Zender, J. F. Kok, X. Liu, Y. Zhang, and S. Albani (2015), Modeling dust as component minerals in the Community Atmosphere Model: development of framework and impact on radiative forcing, *Atmos. Chem. Phys*, *15*, 537–561, doi:10.5194/acp-15-537-2015.

Shao, Y. (2001), A model for mineral dust emission, *J. Geophys. Res.*, *106*(D17), 20,239–20,254, doi:10.1029/2001JD900171.

---

## Author Comment (AC1) · 9 Oct 2016

**Anonymous Referee #1:**

*Major comments:*

*1. Section 2.1: The authors concentrate on a narrow (100km) coastal plain with complex circulation (e.g. sea breezes) but use a model resolution that is too coarse (10km). They use the same resolution also when testing varying emission resolutions. Even if this is due to computational constraints, surely short tests of specific dust events can be conducted at higher resolution to properly test dust dispersion. Can the authors identify such events and complement the paper with higher resolution model runs?*

In this study, we focus on the effect of the spatial resolution of the surface characteristics on the dust generation. It is especially important over the coastal plain where the surface is heterogeneous. We fixed the meteorological forcing to isolate the impact from high-resolution datasets. The text is modified to make this point clearer.
Following the reviewer's advice, we have compared our offline emissions with those calculated in the one-month WRF-Chem simulation of dust storm event in January 2009. The CLM4 results are shown to be in good agreement with WRF-CHEM, producing similar amount and spatial pattern of dust generation. The text is modified to reflect this comparison.

*2. The authors allude to a year-to-year strong variability (Sec.2.3, p.7 l.18), yet only simulate the short period of 3 consecutive years. Are the results meaningful and how are any resulting systematic uncertainties constrained?*

Actually, in this study, we are more concerned to evaluate the effect of spatial changes than temporal variability, which, as we mentioned in the paper, is not very strong and the coastal plane is a steady dust source every year. The phrase on (Sec.2.3, p.7 l.18) is related to flash floods that differ from year to year but have about decadal frequency. The individual flash floods are small-scale events that cannot make significant changes to vegetation and other surface characteristics. They cause short-term local reductions of dust emission due to their impact on soil moisture. In the long run, however, flash floods are responsible for bringing the alluvial material that is a source for dust generation, but this process is important on much longer timescales. Following the comment, we have expanded and improved the text.

*3. The authors do not address dust aging and its potential effects. They should comment if it's relevant before deposition. Does short atmospheric residence time make it not significant?*

We mention the importance of chemical and physical processes that occur during atmospheric transport (dust aging) when discussing dust deposition to the Red Sea. However, the study is aimed at quantifying dust emission processes and does not consider any transport processes. The detailed analysis of dust transport, lifetime and aging is beyond our current scope and requires a separate modeling framework. But, because of the close proximity of the coastal plain to the Red Sea, the transport is short, and atmospheric processing is not very important for dust generated from the coastal plain. Following the comment, we have made this statement clearer.

***4.*** *Section 2.2 describes mostly the dust generation mechanisms that I understand is not the authors own work and is available in the literature. The whole section should be shortened, replacing the details of model parameterisations with references.*

Thank you for pointing it out. Following the suggestion, we have shortened Section 2.2 to make it more focused. However, we want to keep this section, as it is useful to introduce different parameters we use in the paper.

***5.*** *Section 2.4,5: Given the very coarse resolution and the relatively small size and high complexity of the region under study of the MERRAero grid, is it advisable to scale to the emission total? What is the overall point of this section? Sections 2.4 and 2.5 should be merged.*

We paid a special attention to model calibration, as we believe this is one of the key points of the study. The reanalysis products with atmospheric aerosol component have just recently become available. The spatial resolution of MERRAero reanalysis is enough to capture the dust generation hot spot area in the western Arabian Peninsula. Although it is too coarse to represent the spatial structure of dust emission, the integral multi-year estimate of total dust generation from approximately 150000 km$^2$ area is a reasonable reference point for model calibration. Following the reviewer's suggestion, we have merged Section 2.4 and Section 2.5.

***6.*** *Section 3.1: Given the same forcing in all experiments, and differences of a few percent between emission resolutions, what is the point of sensitivity tests? Surely the sensitivity to meteorological conditions and climatic variability is more important and should be studied. The authors should comment and better motivate their study.*

The sensitivity tests are aimed at quantifying the impact of high-resolution satellite datasets on spatial patterns of dust generation. In order to perform these tests, the meteorological forcing should be fixed. Although the integral dust generation does not differ strongly, it does not mean that the impact of input datasets is negligible. Locally, dust emissions may change up to 100 % (Fig. 2). We do not analyze the dust emission long-term variability, as it is beyond the scope of the current paper and might be a topic of the further studies. However, we point out that the spatial pattern of dust hot spots is stable (Fig. 6 a-c, Fig. 7) and thus high-resolution inventories should add value regardless of meteorological forcing. Following the suggestion, we have expanded and clarified the corresponding section.

***7.*** *Section 3.2: Surely statistical testing can't involve changing thresholds until significance is reached. Also are monthly values used when hourly are available? Given the strong diurnal variability, isn't the latter advisable?*

To test the temporal variability of modeled dust emission, we process the hourly values to produce monthly time series of dust frequency and intensity, as seasonal variability was the primary focus of the study. Applying the thresholds for model data cannot be avoided, as the station data also have a "threshold": the visibility and weather code is only reported when it drops below 10000 m. Due to this fact, there is no actual diurnal statistics in visibility data, as reported visibility reductions usually only last for several hours, and diurnal cycle cannot be

validated. Depending on station's location, meteorological and environmental conditions, the same visibility reduction may be caused by the different level of dust loading; therefore, changing thresholds could not be avoided as well. We show that for the most of the stations, the results are statistically significant in the wide range of threshold values. The statistical metric of model spatial performance, however, does not use thresholds.

*8. Section 4: Again given varying vegetation and land type, isn't 3 years to short a time span to produce an emission climatology?*

The focus of the study is on the spatial structure of dust emission with special attention on generation regime in the hot spots. The hot spot areas are the stable structures that are conditioned by land surface characteristics, whereas the amount of generated dust is mostly driven by wind velocity. We agree with the reviewer that wind circulation regime changes, associated with climate variability, lead to changes in dust emission. However, based on results from MERRAero reanalysis we show that interannual variability of dust generation from the coastal plain during $2003 - 2015$ is relatively small ($7.2 \pm 0.5$ Mt a$^{-1}$) and the seasonal cycle is stable (see fig. 7). The three-year simulation period was enough to capture the significant improvement of the dust emission spatial pattern with the use of high-resolution satellite data. We agree with the reviewer, however, that referring to climatological time scale should be replaced by "multi-year estimate".

*Minor comments:*

*1. Using the study-specific acronyms FineALL, HighALL, LowALL, etc. is confusing to the reader. Propose to just change with quoting model resolution and if necessary input data (ie. 1km, 10km, 50km, etc.)*

Following the suggestion, we have changed the experiment acronyms.

*2. Please include the span of years modelled in the abstract.*

The modeling period have been added to the abstract.

*3. Abstract l. 26: appears to be -> is estimated to be [...] suggests -> shows*

Thanks, changed.

*4. P.9 2nd paragraph: First it's stated that it is not rare to have no dust reports, thus the visibility measurement is used, then that the authors prefer sampling as most station visibility are complemented with weather codes. Aren't these statements contradictory? Please re-write the paragraph for more clarity.*

The referred statement means that there were no visibility reduction reports at all, not that visibility reductions came without weather code reports. We have re-formulated these statements.

**5.** *P.9 l.11 current month -> that particular month*

Changed.

**6.** *P.13 l.9 Please elaborate how Yu et al is questioned based on the results of the present study - not just by quoting another paper.*

Thanks for the comment. We have re-formulated the statement to address the questioned issue in more details.

**7.** *P. 19 l.8 Paragraph/line incomplete? Missing full stop?*

Full stop has been added.

**Anonymous Referee #2:**

*1. Abstract, main part, and conclusions: The amount 7:5 Mt a⁻¹ of total dust emission as presented in the abstract, the successively derived magnitude of dust emission from different locations, and the quantification of the amount of iron oxides and phosphorus are all predetermined by the calibration of the model. For the lack of measured dust emission rates, the calibration is done by assuming the same emitted dust amount in the land model as in the MERRAero reanalysis over the investigated time period, which is not based on measured values, but calculated using a dust module.*

This is correct, our total dust generation and other bulk estimates (e.g. mineralogical composition) are based on calibration with respect to the best, in the absence of observations, assimilation product. However, the main focus of this paper is on the spatial heterogeneity and seasonal variability of dust sources from the coastal plain, and their dependence on the surface data sets. These results are invariant with respect to total bulk emission. Our bulk emission estimates could be easily recalibrated when better estimates of total emission are available.

*This approach rests on the assumption that the magnitude of the dust emission in MERRAero is a reliable estimate of the true dust emission from this region. To my knowledge, no evaluation has been published with respect to the dust emission from this region in MERRAero. The authors themselves acknowledge that the resolution of the reanalysis is too low to provide reliable estimates of the dust emission from this region, and they show with their own analysis that the magnitude of dust emission increases with refinement of the horizontal resolution.*

See our answer above. In addition, we have to mention that the MERRAero resolution is too low to resolve the spatial heterogeneity of the emissions, but is less deficient for estimating the bulk emissions from the entire area

*Thus, this suggest the conclusion that the magnitude of the presented total dust emission, the emission amount from individual locations, and the amount of iron oxides and phosphorus are highly uncertain. This uncertainty should be addressed. Perhaps, one could use the variability of the emission in MERRAero from the whole time period 2003-2015, which already has been used in the manuscript by the authors, to provide a first estimate of the lower and upper range of the emission related quantities presented in the paper, especially for the ones presented in the abstract, even though that still wouldn't address any possible bias in the MERRAero dust emission. The issue of the uncertainty and its sources for the estimates provided in the paper should also be thoroughly discussed in the conclusions. Also, when providing the absolute quantities in the abstract, it should be pointed out that the values are just first estimates that are still highly uncertain.*

The uncertainty mentioned by the reviewer is unfortunately, a state of the art problem. However, the proposed approach allows easy recalibration when better estimates are available. We have updated the abstract and the text mentioning that the presented values are the first estimates, and indicated the uncertainty range based on MERRAero variability through the paper, as suggested. We have to mention that although MERRAero dust emission has not been validated yet, the recent paper (Ridley et al., 2016) reports better seasonality of dust AOD in

MERRAero compared to other datasets, and points to potentially better dust emission due to finer spatial resolution and representation of surface winds.

***2. Page 4, lines 10–13:*** *The assumption that the mineral composition of dust aerosols and the mineral composition of the soils in Claquin et al. (1999) and Nickovic et al. (2012) were close does not (always) hold just because of changes during the life cycle of dust from emission to deposition, even more importantly, it does not hold because the measurements of the soil mineral fractions were done for soils that had been wet-sieved. Wet sieving is a technique that strongly disperses soil aggregates (Shao, 2001), which is not realistic for dust emission from the parent soils of the dust sources. This caveat to the assumption made by the authors should be added to the manuscript.*

*Having said this, the authors are mainly interested in the amount of iron oxides and phosphorus. Nickovic et al. assume the same iron oxide fraction in the clay and silt size range, and phosphorus is provided only for the clay and the silt-size range together. Therefore, the fractions of these minerals are less affected by the wet-sieving problem, based on these assumptions. Also, in the present manuscript, only the integrated amount over all size bins defined in the dust module is presented. Thus, other sources of uncertainty probably affect the calculated iron oxide and phosphorus amount more than the wet-sieving problem. The wet-sieving issue still may be are more relevant source of error for the other minerals presented in Figure. 9, though.*

We thank the reviewer for this detailed suggestion. We have updated the text with the remarks on the additional source of uncertainty linked with wet sieving technique.

***3. Page 7, lines 4–7:*** *How the choice for the threshold value for the statistical source function was made should be explained in detail. It is not clear for the reader from simply stating, "The threshold value is chosen with respect to the temporal frequency of the SEVIRI instrument".*

Thanks for pointing this out. The threshold has to be applied to filter out background dust and is usually chosen empirically (Schepanski et al., 2012). We have tested the thresholds in the range of 0.8 – 1.15 and found the spatial patterns of the source functions to be very similar. The chosen threshold value of 1.12 is larger than the one used in (Ginoux et al., 2012) but comparable to other studies (Schepanski et al., 2012). The choice of relatively large threshold was motivated by several reasons. First, the background dust AOD in Arabian Peninsula is much higher than globally observed one. Second, SEVIRI was shown to overestimate AOD under moist conditions and low dust loadings that are the case for the Red Sea coastal plain (Banks et al., 2013). Overall, this larger threshold allows us to better represent intensive dust sources, in contrast, e.g. to (Ginoux et al., 2010; Ginoux et al., 2012) that aimed at capturing and classifying smaller sources. The text has been updated to account for these remarks.

***4. Page 15, line 5****: Do not say "Dust emission climatology", since the analysis is done only for three simulates years. Name the section "Multi-year dust emission" or similar.*

Thanks for this important suggestion, the section has been renamed.

**5. *Page 14, line 23***: *The unit of the total dust emission in the text should be the same as in Fig. 4a, i.e., g m a.*

Changed.

**6. *Page 17, line 29***: *"All of the quantities have a pronounced diurnal cycle, ..." should be phrased more precisely as "Total dust generation, frequency, and maximum emission rate have a pronounced diurnal cycle, ...". The authors themselves discuss the exception for the intensity further below.*

Changed.

**7. *Page 18, line 20***: *Add Scanza et al. (2015).*

The reference has been added.

**8. *Page 31, Table 1***: *The used individual components of the WRF model configuration should be presented in a way that is friendly to the reader who is not an insider of the WRF model. That is, not just by using acronyms, but fully spelled out, with references added and information how these components can be accessed.*

Thanks for the suggestion. The corresponding information has been added.

**References:**

1. Banks, J. R., Brindley, H. E., Flamant, C., Garay, M. J., Hsu, N. C., Kalashnikova, O. V, Klüser, L. and Sayer, A. M.: Intercomparison of satellite dust retrieval products over the west African Sahara during the Fennec campaign in June 2011, Remote Sens. Environ., 136, 99–116, doi:http://dx.doi.org/10.1016/j.rse.2013.05.003, 2013.

2. Ginoux, P., Garbuzov, D., and Hsu, N. C.: Identification of anthropogenic and natural dust sources using Moderate Resolution Imaging Spectroradiometer (MODIS) Deep Blue level 2 data, J. Geophys. Res., 115, DOI:10.1029/2009jd012398, 2010.

3. Ginoux, P., Prospero, J. M., Gill, T. E., Hsu, N. C., and Zhao, M.: Global-scale attribution of anthropogenic and natural dust sources and their emission rates based on MODIS deep blue aerosol products, Rev. Geophys., 50, 1-36, DOI:10.1029/2012RG000388, 2012.

4. Ridley, D. A., Heald, C. L., Kok, J. F. and Zhao, C.: An observationally-constrained estimate of global dust aerosol optical depth, Atmos. Chem. Phys. Discuss., 1–31, doi:10.5194/acp-2016-385, 2016.

5. Schepanski, K., Tegen, I. and Macke, A.: Comparison of satellite based observations of Saharan dust source areas, Remote Sens. Environ., 123, 90–97, doi:10.1016/j.rse.2012.03.019, 2012.

---

## Referee Comment (RC3) · Anonymous Referee #3 · 10 Oct 2016

The authors apply the high-resolution Community Land Model versión-4 (CLM4) together with the Dust Entrainment and Deposition (DEAD) to estimate the dust emissions from the Arabian Red sea coastal plain. The emissions are estimated for a 3-year period (2009-2011) using observed high-resolution satellite land cover and vegetation dataset. The main goal of this study is to generate "new, high-resolution, multi year estimates of spatial and temporal variability of dust emissions". In addition, the authors estimate the mineralogy of the emitted dust by assuming that the composition of the emitted dust is the same as the composition of the soil where it was emitted. The meteorological fields necessary to drive CLM4 are generated with the WRF model at a 10 km x 10 km spatial resolution and the sensitivity of the estimated emitted dust to spatial resolution of the surface characteristics is examined. Results are compared against visibility from station data. The magnitude of the dust emissions is indeed

vey uncertain at present (as mentioned by the authors) and all efforts to reduce these uncertainties are very much welcome. However, I believe that important clarifications and/or modifications need to be done before this work can be published in ACP.

General comments

The main result of the work is not actually quantifying the emissions as suggested by the title and in the manuscript but distributing them in space through the use of the CLM4 model. Authors should be more consistent throughout their paper between their claims and what is actually done.

The total emitted amount is scaled in order to fit the total MERRAero emissions but the scaling factor is never provided. Models do tune their emissions to fit AOD but by doing so they implicitly take into considerations the full aerosol cycle (i.e emission, transport and deposition) and are therefor consistent. However, by simply scaling the emissions to a given model the potential usefulness of the estimate is lost since it is not an independent estimate. It is not clear if it will actually improve performance for other models. How model dependant is this estimate? Even more, how large is the model uncertainty in the emissions for this region? How much dust is emitted by other models in this region? Furthermore, what is the size distribution of the emitted dust in the MERRAero model and how does it compare to the one estimated in this study? Although only the total emission is analysed the size distribution of the emitted dust is key to determine the impact of these emissions in terms of transport and deposition. The authors should provide a discussion addressing these issues.

The authors use visibility data as a mean to validate the estimated flux and draw conclusions on the source of the dust causing this reduced visibility. Visibility is a subjective local measurement reflecting the extinction of light in a given place, but it does not provide any information on the magnitude of the source causing the reduced visibility. Therefore it cannot be concluded on the magnitude of the emission based solely on these observations whether the source is local or not, other variables such as wind

direction and magnitude need to be included for this analysis or a model needs to be applied.

The authors use the Spearman correlation as a statistic to validate the emission intensity. Besides the fact mentioned above that visibility is not appropriate parameter to validate emission intensity, the correlation reflect similarity in variability (spatial and/or temporal) but does not provide any information on the difference or "distance" between the observed variable and the estimated one. The authors should include additional analysis to actually validate the emission intensity.

Specific comments

Page 1, line 26, (Abstract): Remove "The total dust emission from the coastal plain appears to be 7.5 Mt per year". This is not a result of the study but a constrain taken form a model and therefore should not be presented as result.

Page 2, line 31: "Regional uncertainties are probably even higher", on what evidence is this statement based? Authors should provide a reference for this.

Page 3, line 25: "Our principal objective was to obtain new. . .in order to evaluate its impact on the Red Sea". This objective should be reformulated and made consistent with the actual work done in this study. The emissions are first of all not estimated since they are scaled and for the same reason they can't be new. Furthermore, the impact of the dust deposition on the Red Sea is not evaluated. The work as presented does not have the tools to address this issue. I would therefore strongly recommend removing this last part or reformulating it in order to make it consistent with the work that is presented in the manuscript.

Page 4, line 4: Replace "availably" with "availability".

Page 4, line 10: ". . .are close of those of the parent soil." Later in the text it is said that they are the same, what is it? The same or close? Please be consistent.

Page 5, lines 3-7: How was the setup or configuration of the WRF model defined?

Please specify.

Page 6, line 10: The variable "S" should be presented as source function at this point and not on line 14 as it is at present.

Page 7, line 2-3: Please provide a reference for the assumption that the intensity of the dust source is proportional to the frequency of occurrence of atmospheric dust. On what is this based?

Page 7, lines 6-7: It is still unclear how the threshold of 1.12 was chosen. Please elaborate.

Page 9, line 3: Why these two thresholds? Please explain why these two thresholds were used. Page 10, lines 5-8: Please provide a reference for what is said in these lines.

Page 10, line 15: According to whom is it not captured?

Page 12, lines 30-31: "Yu et al. (2013) offered several explanations for this". It is not clear to what does it refer. One would expect it refers to the previous statement, but then on the next sentence satellite data are mentioned. Please reformulate.

Page 13, lines 5-9: I do not agree with what the authors claim in these lines. Whether the data used in this study nor the analysis conducted allow to conclude on whether the dust is emitted locally or transported from elsewhere. High correlations only indicate similar variability but are not an indication of distance between observations and model. One could have high correlations but also have dust coming from elsewhere. The explanation may appear reasonable, but it is not supported (nor refuted) by evidence presented in the manuscript. I suggest either removing completely these lines or reformulating it presenting evidence to support this claim.

Page 13, line 25: Is this model skill the correlation coefficient? Or does it refer to another statistic? Please clarify.

Page 14, line 7: "provide quite realistic results", please reformulate. How much is "quite"? Please explain better why only the FineALL case is only consider in the remaining analysis.

Page 15, line 30: Replace or eliminate "reasonably". How much is "reasonably"?

Page 15, line 31: Although SM1 and SM2 can be identified in MERRAero, the authors should acknowledge the differences between both representations (this work and MERRAero). For instance MERRAero locates a dust source further to the north than suggested by this study.

Page 17, line 1: I do not fully agree on the statement made on the first sentence of the paragraph. Although hotspots present variability consistent with the seasonal cycle, not all features can be explained by the hot spots (hotspots show very little variability from March to August in contrast to emissions from the entire region which shows strong seasonality). The seasonal cycle of sources other than hotspots should also be included in the figure to clarify the real weight of hotspots in modulating the emissions in the area of interest.

Page 17, lines 29-30: "All quantities...", this is actually not entirely true since figure 8b presents variability not consistent with the solar peak and this is actually described later on. Please make the analysis consistent.

Page 18, lines 15-18: Why is so little said about the diurnal cycle of the dust maximum emission? Or why is it included? Authors should spend at least the same effort in analysing it as on the other variables, otherwise I would suggest removing it. Actually, how does it contribute to the general goal of this study?

Page 19, line 1: I would suggest include "estimated" or "calculated" before "emitted mineral fraction".

Page 19, lines 17-20: This entire paragraph should be removed from this section (it is not a conclusion of this work) and placed after the last paragraph of section 2.1.
**Interactive comment**

Page 19, lines 24-26: "The results confirmed…." This conclusion cannot be made based on the evidence presented in this work. See comment made before.

Page 19, lines 27-28: This is true for the case when source function is used, while when the source function is not used this is not the case as stated in lines 25-28 of page 13. Please reformulate in order to make it consistent.

Page 20, line 23: Shouldn't it be early afternoon when referring to 12:00-14:00 UTC?

Page 20, lines 28-31: First of all the 7.5 Mt/a are not estimated but imposed. This should be corrected. Then, the fact that emissions and deposition have comparable magnitude does not allow to conclude that it is an essential source of nutrients for the Red Sea, specially if one considers that the total amount was imposed from the beginning. Although one would expect that some of the emitted dust in the coastal plain should be deposited in the Red Sea, how much of it needs to be determined by another study. I would suggest removing this sentence.

---

## Author Comment (AC2) · 19 Oct 2016

**Anonymous Referee #3:**

**General comments:**

**1.** The main result of the work is not actually quantifying the emissions as suggested by the title and in the manuscript but distributing them in space through the use of the CLM4 model. Authors should be more consistent throughout their paper between their claims and what is actually done.

In this study we quantified the spatial distribution of dust emissions in the Red Sea coastal plain. The total emission estimate is obtained from the MERRAero reanalysis and we specifically emphasized this in the text. This figure is calculated by us and was previously not known. Therefore we believe it is legitimate to use the term quantification.

2. The total emitted amount is scaled in order to fit the total MERRAero emissions but the scaling factor is never provided.

The scaling coefficient is provided in Table 4. We have stressed the reference in the text to make it more noticeable.

**3.** Models do tune their emissions to fit AOD but by doing so they implicitly take into considerations the full aerosol cycle (i.e emission, transport and deposition) and are therefor consistent. However, by simply scaling the emissions to a given model the potential usefulness of the estimate is lost since it is not an independent estimate.

The reviewer is, probably, familiar with the fact that the closure of the dust mass budget requires "... to quantify precisely the amount of emitted dust, the atmospheric dust load, and the deposited mass (dry and wet). However, this budget is sufficiently constrained if, at least, two of these three terms are quantified ... " (Bergametti and Forêt, 2014). Therefore, the models that tune their emissions to fit AOD (i.e. dust burden) also incorporate uncertainty and cannot provide an "independent" emission estimate. So we question the remark that this approach is more consistent if the study is aimed at assessing dust emission mass. The scaling based on AOD observations alone could still be biased due to other reasons, i.e. errors in deposition velocity or size distribution representation.

In our modeling framework, we calculate only dust emissions, scale them using the most recent and reliable reanalysis dataset (see the answer to the next remark) and verify their spatial and temporal variability using station visibility observation. To our best knowledge, this is the most consistent and straightforward approach possible at present.

**4.** It is not clear if it will actually improve performance for other models. How model dependant is this estimate? Even more, how large is the model uncertainty in the emissions for this region? How much dust is emitted by other models in this region?

We are not sure if we understand the question completely. In our study, we did not compare different models, but different land cover dataset. Our results show that, certainly, using finer land cover datasets will improve spatial variability of the dust emissions and will be beneficial for the models.

The uncertainties mentioned by the reviewer are a state of the art problem. To our best knowledge, only a few estimates of dust emission made so far for the Arabian Peninsula, and most of them are done using coarse-resolution models. No specific work has been done for the Red Sea coastal plain, so our work is a pioneering attempt of this type.

The MERRAero meteorology-aerosol reanalysis is the most appropriate data source to tune the regional dust emissions. E.g., Ridley et al., (2016) reported better seasonality of dust AOD in MERRAero compared to other datasets and pointed to potentially better dust emission due to finer spatial resolution and representation of surface winds. However, we agree with the reviewer that the uncertainty estimation needs additional attention. Following the suggestion from the anonymous reviewer #2, we have complemented the manuscript with the uncertainty range estimation based on the interannual variability of dust emission in MERRAero reanalysis. Moreover, we have performed the one-month WRF-Chem simulation of dust storm event in January 2009 to compare with our off-line emissions from this study. The CLM4 results appear to be in good agreement with WRF-Chem, producing similar intensity and spatial pattern of dust emission. The text is modified to reflect this comparison.

5. Furthermore, what is the size distribution of the emitted dust in the MERRAero model and how does it compare to the one estimated in this study? Although only the total emission is analysed the size distribution of the emitted dust is key to determine the impact of these emissions in terms of transport and deposition. The authors should provide a discussion addressing these issues.

It is well known that dust size distribution is important for dust transport and deposition. However, in the current study, we only simulate dust generation and do not consider transport and deposition. Within this framework, analyzing the size distribution of emitted dust would not allow to reduce uncertainty or make any additional quantitative conclusions on its transport and deposition. To do it, one needs to simulate the full cycle of airborne dust, which is a subject of future research.

6. The authors use visibility data as a mean to validate the estimated flux and draw conclusions on the source of the dust causing this reduced visibility. Visibility is a subjective local measurement reflecting the extinction of light in a given place, but it does not provide any information on the magnitude of the source causing the reduced visibility. Therefore it cannot be concluded on the magnitude of the emission based solely on these observations whether the source is local or not, other variables such as wind direction and magnitude need to be included for this analysis or a model needs to be applied.

The authors use the Spearman correlation as a statistic to validate the emission intensity. Besides the fact mentioned above that visibility is not appropriate parameter to validate emission intensity, the correlation reflect similarity in variability (spatial and/or temporal) but does not provide any information on the difference or "distance" between the observed variable and the estimated one. The authors should include additional analysis to actually validate the emission intensity.

The reviewer, probably, refers to weather code reports when talking about the subjective character of the measurements. The visibility measurements that complement the weather code report are not subjective, as they are usually done by ASOS (Automated Surface Observing System) visibility sensor. We agree with the reviewer on his concerns regarding the limitations of visibility and weather code data. Indeed, the detailed discussions about the limitation are already present in the manuscript. However, we cannot agree that visibility measurements are not appropriate for testing dust emission models. In the absence of direct observation of emission, visibility data are the most relevant data sources for these purposes. These observations provide valuable information and may serve as a reference for qualitative comparison with modeled dust emission fluxes and determine optimal model configuration (Engelstaedter et al., 2006; Tegen, 2003). They were used in a large number of dust-related studies. For example, the present weather code reports from meteorological records have been used for evaluation of dust event frequency and dust climatology (Goudie and Middleton, 2006; Shao and Dong, 2006; Wang et al., 2011; Notaro et al., 2013; Yu et al., 2013; Cowie et al., 2014; Hamidi et al., 2014), and derive soil erodibility fields (Shao, 2008). In (Camino et al., 2015; Rezazadeh et al., 2013; Shao et al., 2003) parameterization for assessing near-surface dust concentration from visibility measurement has been proposed. Mahowald et al. (2007) stated that visibility-derived observations should better capture the temporal variability of surface dust fluxes compared to AOD measurements. In our study, we use both weather code reports and visibility measurements to evaluate the frequency and intensity of simulated dust emission.

**Specific comments:**

**1.** Page 1, line 26, (Abstract): Remove "The total dust emission from the coastal plain appears to be 7.5 Mt per year". This is not a result of the study but a constrain taken form a model and therefore should not be presented as result.

Following the reviewer's comment, we have reformulated this phrase to make clear we obtained this figure from the reanalysis. This was first time calculated so it is a legitimate result of all reasonable means. There are tons of results in the literature obtained from the reanalysis data and nobody question their originality based only on that they are obtained from a reanalysis. The total dust emission estimate from the coastal plain is important for this study and, we believe, has to be clearly outlined in the abstract.

**2.** Page 2, line 31: "Regional uncertainties are probably even higher", on what evidence is this statement based? Authors should provide a reference for this.**

It could be proofed straight mathematically, as the integral of the function over the entire globe is less variable than a function itself. Huneeus et al. (2011) reported that globally averaged model estimates of dust emission, deposition and optical properties vary by a factor of 10. Apparently, these discrepancies are driven from even larger regional ones, as global models do

not simulate regional processes. Following the suggestion, we improved the text to make it more clear.

**3.** Page 3, line 25: "Our principal objective was to obtain new. . .in order to evaluate its impact on the Red Sea". This objective should be reformulated and made consistent with the actual work done in this study. The emissions are first of all not estimated since they are scaled and for the same reason they can't be new. Furthermore, the impact of the dust deposition on the Red Sea is not evaluated. The work as presented does not have the tools to address this issue. I would therefore strongly recommend removing this last part or reformulating it in order to make it consistent with the work that is presented in the manuscript.

We should say that the study has been motivated by that the coastal emissions are important for the Red Sea as a significant amount of this material could deposit to the Sea. We agree with the reviewer that as long as the impact on the Red Sea is not calculated directly, the statement should be removed and the objectives to be re-formulated.

4. Page 4, line 4: Replace "availably" with "availability".

Thanks, replaced.

5. Page 4, line 10: "... are close of those of the parent soil." Later in the text it is said that they are the same, what is it? The same or close? Please be consistent.

They are the same as in the parent soil. Changed.

**6.** Page 5, lines 3-7: How was the setup or configuration of the WRF model defined? Please specify.

WRF setup generally follows default recommendations from the user guide and is identical to that used in (Jiang et al., 2011). Following the suggestion, we have included this information in the text.

7. Page 6, line 10: The variable "S" should be presented as source function at this point and not on line 14 as it is at present.

Thanks, changed.

8. Page 7, line 2-3: Please provide a reference for the assumption that the intensity of the dust source is proportional to the frequency of occurrence of atmospheric dust. On what is this based?

The "frequency method" was first proposed by Prospero et al. (2002), and later used in a number of other studies (Ginoux et al., 2010, Ginoux et al., 2012, Schepanski et al., 2012). Following the suggestion, the text was updated with the references.

9. Page 7, lines 6-7: It is still unclear how the threshold of 1.12 was chosen. Please elaborate.

The choice of this threshold has been already explained in the reply to the comment #3 by anonymous referee #2.

**10. Page 9, line 3: Why these two thresholds? Please explain why these two thresholds were used.**

Please refer to the question #7 by anonymous referee #1. Depending on weather station's location, meteorological and environmental conditions, the same visibility reduction may be caused by the different level of dust loading; therefore, changing thresholds could not be avoided. These two particular thresholds were chosen empirically and are aimed to demonstrate that results are not very sensitive to a threshold value.

**11. Page 10, lines 5-8: Please provide a reference for what is said in these lines.**

Following the suggestion, the GOCART aerosol scheme description paper was repeatedly referenced in this place of the text.

**12. Page 10, line 15: According to whom is it not captured?**

We have also analyzed the dust emissions in the MACC reanalysis. We have updated the text to make it clearer.

13. Page 12, lines 30-31: "Yu et al. (2013) offered several explanations for this". It is not clear to what does it refer. One would expect it refers to the previous statement, but then on the next sentence satellite data are mentioned. Please reformulate.

Thanks for the suggestion. The statement is related to issues discussed in author's own paper. The phrase was reformulated.

14. Page 13, lines 5-9: I do not agree with what the authors claim in these lines. Whether the data used in this study nor the analysis conducted allow to conclude on whether the dust is emitted locally or transported from elsewhere. High correlations only indicate similar variability but are not an indication of distance between observations and model. One could have high correlations but also have dust coming from elsewhere. The explanation may appear reasonable, but it is not supported (nor refuted) by evidence presented in the manuscript. I suggest either removing completely these lines or reformulating it presenting evidence to support this claim.

It is not clear enough from the reviewer's remark what particular point of our claims is questioned. Following that, we have expanded the corresponding section to make the discussion clearer. The idea that dust activities in our area of interest have small spatial scales was proposed by Yu et al. (2013). The authors reported low correlations (0.1 - 0.3) between the monthly AOD observations and station dust reports in the west of Arabian Peninsula, compared to much higher ones in the central and eastern Peninsula (usually more than 0.4). This means that, for some reason, dust events reported on stations could not be detected by satellite

instruments. The authors also report that there is a large probability of observing low AOD values on dusty days. Several explanations for this contrast were proposed. Noting the shortcomings of remote sensing instruments that perform worse in the complex mountainous terrain of the western Arabian Peninsula, they claimed that this contrast might be caused by the small spatial-temporal scales of dust processes. In our manuscript, we claim our results to support these ideas and to be consistent with the proposed mechanisms. Higher correlation coefficients of station dust event time series and simulated emission fluxes compared to those reported by Yu et al. (2013) suggest that a large part of detected variability could be explained by local dust generation. Mahowald et al. (2007) also supported this conclusion suggesting that station observations should better capture the temporal variability of surface dust fluxes compared to AOD measurements.

**15.** Page 13, line 25: Is this model skill the correlation coefficient? Or does it refer to another statistic? Please clarify.

Thanks for noting that, we meant the correlation coefficient here. Changed.

**16.** Page 14, line 7: "provide quite realistic results", please reformulate. How much is "quite"? Please explain better why only the FineALL case is only consider in the remaining analysis.

The FineALL experiment was used in the remaining analysis as it has the highest resolution and the spatial correlation for it is the highest.

17. Page 15, line 30: Replace or eliminate "reasonably". How much is "reasonably"?

We consider the correspondence of dust emission patterns between the two datasets as reasonable, with regard to the coarse resolution of MERRAero. Most of the hot spot areas that are present in MERRAero are also present in CLM4. CLM4 also features smaller hot spots, that could not be resolved in MERRAero.

18. Page 15, line 31: Although SM1 and SM2 can be identified in MERRAero, the authors should acknowledge the differences between both representations (this work and MERRAero). For instance MERRAero locates a dust source further to the north than suggested by this study.

Thanks for this suggestion. Following this and the previous comment, we have updated the text with a more detailed discussion about the location of emission hot spots in our run and MERRAero reanalysis.

**19.** Page 17, line 1: I do not fully agree on the statement made on the first sentence of the paragraph. Although hotspots present variability consistent with the seasonal cycle, not all features can be explained by the hot spots (hotspots show very little variability from March to August in contrast to emissions from the entire region which shows strong seasonality). The seasonal cycle of sources other than hotspots should also be included in the figure to clarify the real weight of hotspots in modulating the emissions in the area of interest.

We thank the reviewer for this thoughtful suggestion. We agree that the corresponding statement is not fully correct. We have revised the paragraph and reformulated our claim.

**20.** Page 17, lines 29-30: "All quantities. . .", this is actually not entirely true since figure 8b presents variability not consistent with the solar peak and this is actually described later on. Please make the analysis consistent.

Thanks for this suggestion. We have made the statement more clear.

**21.** Page 18, lines 15-18: Why is so little said about the diurnal cycle of the dust maximum emission? Or why is it included? Authors should spend at least the same effort in analysing it as on the other variables, otherwise I would suggest removing it. Actually, how does it contribute to the general goal of this study?

We consider the maximum emission rate an important characteristic that provides the reader with a better understanding of the diurnal cycle of dust generation. Therefore, we prefer to retain the corresponding figure. Following the suggestion, we have expanded the manuscript with a more detailed discussion of the diurnal cycle of maximum dust emission rate.

**22.** Page 19, line 1: I would suggest include "estimated" or "calculated" before "emitted mineral fraction".

Thanks for the suggestion, changed.

**23.** Page 19, lines 17-20: This entire paragraph should be removed from this section (it is not a conclusion of this work) and placed after the last paragraph of section 2.1.

Following the suggestion, the paragraph has been shifted.

**24.** Page 19, lines 24-26: "The results confirmed. . .." This conclusion cannot be made based on the evidence presented in this work. See comment made before.

Following our comment above, we suppose this conclusion should be retained.

**25.** Page 19, lines 27-28: This is true for the case when source function is used, while when the source function is not used this is not the case as stated in lines 25-28 of page 13. Please reformulate in order to make it consistent.

Thanks for the important suggestion. The statement has been reformulated.

26. Page 20, line 23: Shouldn't it be early afternoon when referring to 12:00-14:00 UTC?

Thanks for spotting. Changed.

**27.** Page 20, lines 28-31: First of all the 7.5 Mt/a are not estimated but imposed. This should be corrected. Then, the fact that emissions and deposition have comparable magnitude does not

allow to conclude that it is an essential source of nutrients for the Red Sea, specially if one considers that the total amount was imposed from the beginning. Although one would expect that some of the emitted dust in the coastal plain should be deposited in the Red Sea, how much of it needs to be determined by another study. I would suggest removing this sentence.

Thanks for the suggestion. These few statements have been changed to reflect the reviewer's concerns.

**References:**

[revised manuscript text omitted]